# Baltic Sea freshwater content

Urmas Raudsepp, Ilja Maljutenko, Amirhossein Barzandeh, Rivo Uiboupin, Priidik Lagemaa

Department of Marine Systems, Tallinn University of Technology, Tallinn, 12618, Estonia
*Correspondence to*: Ilja Maljutenko (ilja.maljutenko@taltech.ee)

**Abstract.** The Baltic Sea is a brackish shallow sea, the state of which is determined by the mixing of freshwater from net precipitation and runoff with the salty water from the North Sea inflows. The freshwater content (FWC) of the Baltic Sea is calculated from the Copernicus regional reanalysis data covering the period 1993–2021. The FWC in the Baltic Sea shows a steady decrease over the past two decades with a linear trend of 23.9 km$^3$ per year; however, the trend has significant spatial

variability. The Gulf of Bothnia has a positive FWC tendency, while the Baltic proper has a negative FWC tendency. Temporal changes of FWC are opposite between the Bothnian Bay in the north and the southern Baltic proper. In the Bothnian Bay, interannual changes of FWC are positively correlated with river runoff and net precipitation and negatively correlated with salt transport. In the southern Baltic proper, the variations of FWC and salt transport through the Danish straits are negatively correlated from 1993 until 2010 but positively correlated thereafter. The seasonal freshwater content

reflects the specific hydrophysical conditions of each sub-basin, with northern basins being influenced by seasonal river runoff and ice formation and melting, while the southern basins are more responsive to subsurface salinity changes due to salt transport through the Danish straits.

*Keywords*: freshwater content, reanalysis data, river runoff, net precipitation, salt transport, Baltic Sea

**1 Introduction**

Climate warming has resulted in the intensification of the global hydrological cycle but not necessarily on the regional scale (Pratap and Markonis, 2022). The increase of net precipitation over land and sea areas, decrease of ice cover, and increase of river runoff are the main components of the global hydrological cycle that increase freshwater content (FWC) in the ocean (Boyer et al., 2007) and decrease ocean salinity. All the components can be directly estimated but might have significant

uncertainties. Instead, the ocean salinity change can be used as a marker of the water cycle change (Durack et al., 2012).

In the case of an open part of the ocean, e.g. a regional sea, using salinity as proxy for FWC includes an additional blurring aspect, which is water transport through the open boundaries between the basin under consideration and its surrounding area. The impact of water exchange on the changes of FWC is significant if not dominant. In that case, changes of FWC may not represent the actual changes of freshwater input from the above-mentioned sources.

The Baltic Sea is one of the marginal seas where water salinity and FWC are strongly influenced by the water exchange with the North Sea. The Major Baltic Inflows (MBIs) are the most voluminous event-type sources of saline water to the Baltic

Sea (Mohrholz, 2018). The frequency and intensity of the MBIs and other large volume inflows have no long-term trends but do have a multidecadal variability of about 30 years (Mohrholz, 2018; Lehmann and Post, 2015; Lehmann et al., 2017; Radtke et al., 2020). Smaller barotropic and baroclinically driven inflows transport saline water into the halocline or below

it, depending on the density of the inflow water (Reissmann et al., 2009). The inflows of saline water are forced by winds from the west and outflows by winds from the east.

Direct total input of freshwater to the Baltic Sea consists of river runoff and net precipitation. The total river runoff from the Baltic Sea catchment area shows no statistically significant trend but a variability of about 30 years (Meier et al., 2019a, b) and a pronounced decadal variability of accumulated anomaly of runoff (Lehmann et al. ,2022). The variations in runoff

explain about 50% of the long-term variability in volume-averaged Baltic Sea salinity (Meier and Kauker, 2003; Lehmann et al., 2022), while the direct dilution of the Baltic Sea water by freshwater accounts for 27% of the interannual variations (Radtke et al., 2020). During 1950-2018, precipitation averaged over the Baltic Sea catchment area had a trend of 1.44 mm year$^{-1}$ (Meier et al., 2022).

Thus, the long-term salinity of the Baltic Sea is determined by saline water inflows from the North Sea (wind forcing) and its

dilution with freshwater originating from numerous rivers across the Baltic coast and net precipitation (Winsor et al., 2001; Meier and Kauker, 2003; Gustafsson and Omstedt, 2009; Schimanke and Meier, 2016; Lehmann et al., 2022). There is no clear long-term trend of the mean salinity of the Baltic Sea, but there are multidecadal oscillations of about 30 years (Kniebusch et al., 2019). A 30-year variability has been found for the salinity, river runoff, and saltwater inflows (Radtke et al., 2020). The Baltic Sea salinity also has a natural centennial variability (Kniebusch et al., 2019).

A specific feature of the Baltic Sea is the large difference in sea surface salinity, ranging from about 20 g kg$^{-1}$ in the Kattegat to 2 g kg$^{-1}$ in the Bothnian Bay (Leppäranta and Myrberg, 2009). Kniebusch et al. (2019) found a positive trend of centennial changes in the north-south gradient of the surface salinity and river runoff in the northern catchment area. Multi-decadal oscillations control the long-term variations of surface salinity and its meridional gradient with a period of about 30 years (Radtke et al., 2020).

A common approach is to use salinity to describe the energy and water cycles in the Baltic Sea (Lehmann et al., 2022; Meier et al., 2022). In this study, instead of using spatially mean salinity of the Baltic Sea, we suggest the concept of FWC (Boyer et al., 2007) for the description of the physical state of the Baltic Sea. Previously, a concept of FWC has been used to estimate the freshwater budget of the Baltic Sea (Winsor et al., 2001) and for the geographical spreading of spring-time river runoff (Eilola and Stigebrandt, 1998).

The aim of this study is to analyse the changes of the Baltic Sea FWC during the period of 1993–2021. The MBI in 1993 ended the stagnation period with no MBIs that lasted for about 10 years (1983–1993). During the stagnation period salinity was below average, stratification weakened, and hypoxic area decreased (Lehmann et al., 2022). The period of 1993–2021 includes the third in volume MBI in 2014 (Mohrholz et al., 2015) and several of the other barotropic large volume inflows (Mohrholz, 2018). We focus on the changes of the FWC in the whole Baltic Sea as well as its sub-basins. We investigate the

trends in FWC and observe its seasonal changes. A qualitative explanation of the physical processes behind the dynamics of FWC is provided.

## 2 Data and methods

The BALMFC CMEMS reanalysis product (data ref. 1, Table 1) is calculated using the Nemo-Nordic 1.0 ocean model (Hordoir et al., 2019). The horizontal resolution of the model is approximately 2 nautical miles, and there are 56 vertical

levels. Vertical resolution varies from 3 m at the surface to 10 m below the 100 m depth. The model without data assimilation has been thoroughly validated (Hordoir et al., 2019). The Copernicus model system uses the Localised Singular Evolutive Interpolated Kalman filter data assimilation method (Liu and Fu, 2018). A detailed quality assessment of the reanalysis product (data ref. 1, Table 1), using K-means clustering algorithm (Raudsepp and Maljutenko, 2022), is provided in Appendix A.

**Table 1. CMEMS and non-CMEMS products used in this study, including information on data documentation.**

| Product ref. no. | Product ID & type | Data access | Documentation |
|---|---|---|---|
| 1 | BALTICSEA_REANALYSIS_PHY_003_011; Numerical models | EU Copernicus Marine Service Product, (2021) | Quality Information Document (QUID): Liu et al. (2019) Product User Manual (PUM): Axell et al. (2021) |
| 2 | ERA5; Numerical models | Copernicus Climate Change Service (C3S) (2023) | Hersbach et al. (2023) |
| 3 | IOW-THREDDS-BMIP_bmip_rivers_2019-10-10-1; Numerical models and Observations | IOW THREDDS (2019) | Väli et al. (2019) |

The FWC is calculated according to Boyer et al. (2007)

$$FWC = \frac{\rho(Sref,Tref,p)}{\rho(0,Tref,p)} \frac{S_{ref}-S}{S},$$ (1)

where $S(x, y, z, t)$ and $S_{ref}(x, y, z)$ are actual salinity and reference salinity, respectively, and $x,y,z,t$ are zonal, meridional,

vertical and temporal coordinates, respectively. The density, $\rho$, is calculated according to the TEOS10 (IOC et al., 2010). The key issue of FWC calculations lies in how the reference salinity is defined. The climatological range of salinity in the Baltic Sea varies from the freshwater conditions in the northern and eastern parts to the oceanic water conditions in the Kattegat. We follow the Boyer et al. (2007) formulation and calculate the climatological FWC from the three-dimensional temperature ($Tref$) and salinity ($Sref$) fields averaged over the period of 1993–2020.

The other widely used formulation of FWC is as follows (e.g. Gustafsson and Stigebrandt, 1996):

$$FWC = \frac{\rho(Sref,Tref,p)}{\rho(0,Tref,p)} \frac{Sref-S}{Sref}.$$ (2)

Both formulations are derived from the concept of the mixing of two water masses with different salinities by using conservation of salt but have different mechanistic approaches, which are explained in detail in Appendix B.

The total volume of freshwater needed to dilute the water with salinity $S_{ref}$ to the salinity $S$, if $S<S_{ref}$ or should be removed to obtain water with the salinity $S$, if $S>S_{ref}$. is

$$FWC(t) = \iiint_V FWC(x,y,z,t)dxdydz \ [\text{m}^3 \ \text{m}^{-3}].$$

A vertical distribution of the FWC is calculated as

$$FWC(z,t) = \iint_A FWC(x,y,z,t)dxdy \ [\text{m}^2 \ \text{m}^{-3}],$$

and horizontal distribution of the FWC is calculated as

$$FWC(x,y,t) = \int_D FWC(x,y,z,t)dz \ [\text{m} \ \text{m}^{-3}].$$

The $V$ and $A$ correspond to the volume and area of the Baltic Sea or its sub-region, as shown on Fig 1. The $D$ corresponds to depth from surface to bottom at a specific location.

The sea ice volume, $V_i$, is calculated from the same BALMFC CMEMS reanalysis product (data ref 1, Table 1), based on the LIM3 model configuration (Pemberton et al., 2017). The $V_i$ is calculated for each model grid cell (x,y) using total ice thickness, $H_i$, and ice concentration, $C_i$:

$$V_i(x,y,t) = H_i(x,y,t) * C_i(x,y,t) * dA(x,y)$$

where $dA$ is the area of each grid-cell.

Hourly precipitation and evaporation data have been extracted from the ERA5 reanalysis (data ref 2, Table 1) from the period of 1993–2020. Net precipitation was calculated by subtracting evaporation from precipitation. Thereafter, the net precipitation was interpolated onto a reanalysis model grid, and total net precipitation was estimated for the wet grid-cells of each sub-basin (Fig. 1). The net precipitation anomalies were calculated relative to the period 1993–2020.

The total runoff from the Baltic Sea rivers was estimated from the river discharge database (data ref 3, Table 1) of the Baltic Model Intercomparison Project (Gröger et al., 2022). The runoff to each sub-basin was calculated by summing the runoffs from each river discharging to the corresponding sub-basin (Fig. 1). The runoff data covered the period 1993–2018, and the anomalies were calculated relative to the same reference period.

Salt transport was estimated by calculating the salt flux at the boundaries of each sub-basin (Fig. 1 for the location of the transects). Daily salt transport through each transect was calculated as a salinity and perpendicular velocity product. The annual mean salt transports were calculated by averaging the transects of daily transports over each year and later integrated over the vertical and corresponding horizontal dimension. The positive direction is determined according to the estuarine transport definition, i.e. inflow is from the ocean and outflow is from the head of the estuary.

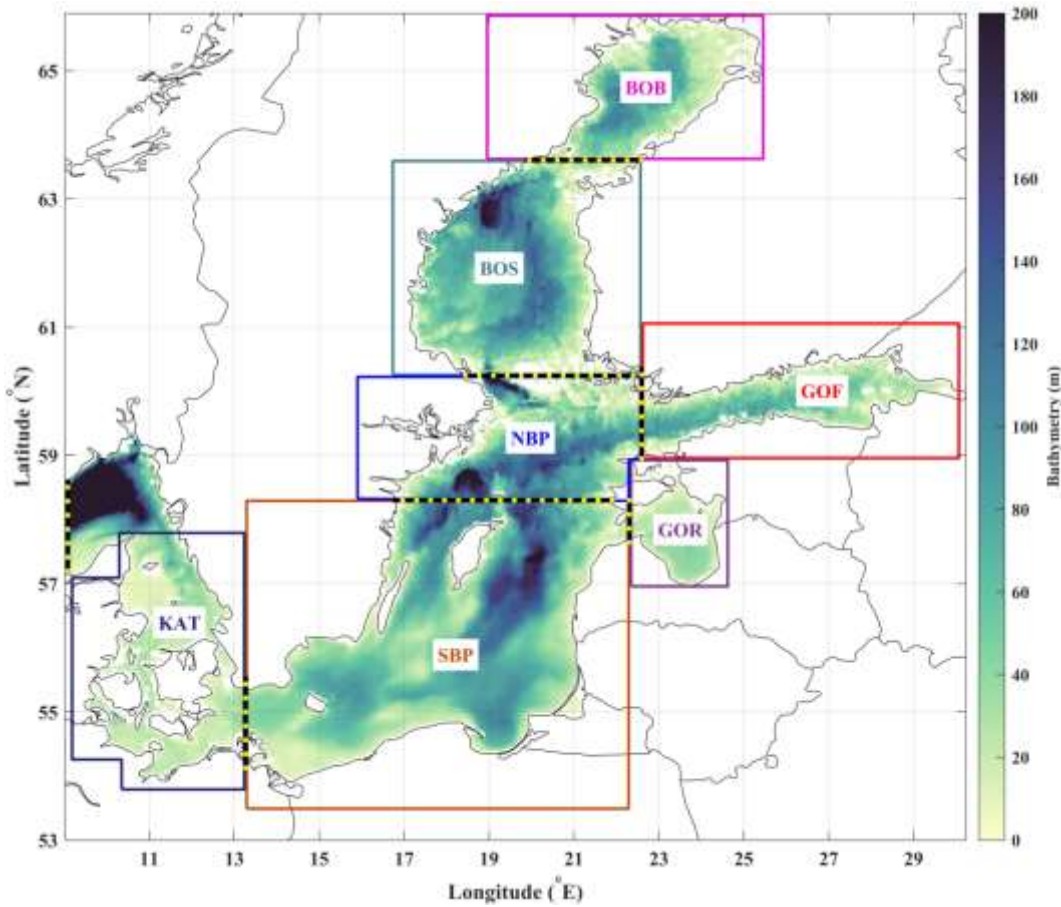

**Figure 1. Map of the Baltic Sea depth distribution (data ref. 1., Table 1). Boxes indicate the boundaries used to calculate the freshwater content for different sub-basins. The transects used to calculate salt transport between sub-basins are represented by yellow-black dash-lines. The abbreviations for the sub-basins are as follows: KAT (Kattegat), SBP (Southern Baltic Proper), NBP (Northern Baltic Proper), BOS (Bothnian Sea), BOB (Bay of Bothnia), GOF (Gulf of Finland), GOR (Gulf of Riga).**

## 3 Results

Time series of the FWC and linear trends of the Baltic Sea and its sub-basins are presented in Fig. 2. Even if the calculated trends are not statistically significant, they provide information about the tendency of FWC changes. The FWC of the Baltic Sea has a negative trend of $-23.9 \pm 0.7$ km$^3$ y$^{-1}$ ($p < 10^{-3}$) superimposed by irregular decadal variations (Fig. 2a). The trends are variable over the whole Baltic Sea (Fig. 2). It changes sign from positive in the northern sub-basins to neutral in the eastern sub-basins and to negative in the central and southern sub-basins (Fig. 2). The decrease of the FWC in the southern Baltic Proper (Fig. 2) contributes the most to the overall decreasing trend in the Baltic Sea. If we look at the continuous distribution of the trends in the Baltic Sea, we see opposite temporal regimes of the FWC in the Bothnian Bay and in the

Baltic Proper, with the Bothnian Sea being the transition area (Fig. 3). Although there is no trend in the Gulf of Finland as a whole (Fig. 2), the eastern part has a small negative tendency, while the western part shows a small positive tendency. The shallow Gulf of Riga has a negligible trend. The trends vanish in the southwestern Baltic Sea and in the Kattegat area (Fig. 3).

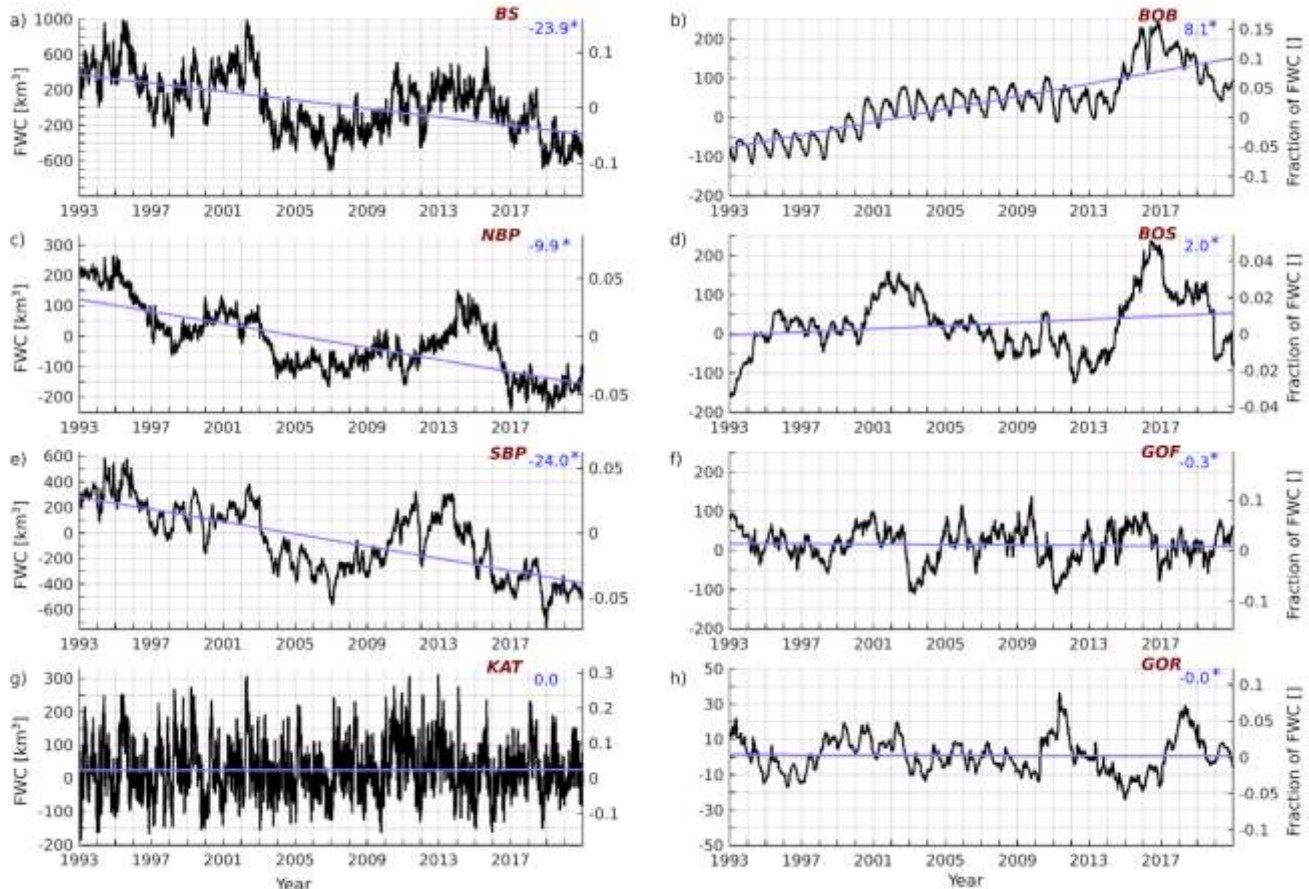

**Figure 2. Freshwater content time series in the Baltic Sea (a) and in different sub-basins (b-h). The trend of FWC in the corresponding basin is shown in the upper right corner (km$^3$ year$^{-1}$, asterisk p>0.05) and plot using the blue line. The abbreviations for the sub-basins are as follows: BS (Baltic Sea), NBP (Northern Baltic Proper), SBP (Southern Baltic Proper), KAT (Kattegat), BOB (Bay of Bothnia), BOS (Bothnian Sea), GOF (Gulf of Finland), GOR (Gulf of Riga). (data ref 1, Table 1 )**

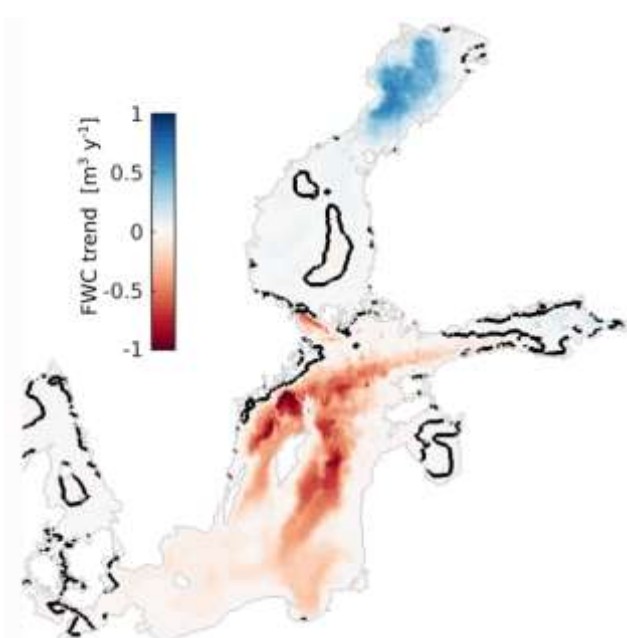


**Figure 3. Spatial distribution of depth normalized FWC trends, with zero-isoline highlighted in black.**

The spatial distribution of the trends indicates a possible coherence of salinity dynamics in different sub-basins, which is checked further on by calculating correlation coefficients between the basins. The correlation coefficients calculated pairwise between detrended FWC time series (Table 2) show a high positive value between the southern and northern Baltic

proper (R=0.8) and between the Bothnian Bay and the Bothnian Sea (R=0.6), while the correlation between the Bothnian Bay and southern and northern Baltic proper is negative (R=-0.6). This suggests that although temporal variability is opposite, the dynamics of FWC are linked over the Baltic Sea.

**Table 2. Correlations table of the FWC between the sub-basins (Fig. 1) of the Baltic Sea (data ref. 1, Table 1). The abbreviations for the sub-basins are as follows: KAT (Kattegat), SBP (Southern Baltic Proper), NBP (Northern Baltic Proper), BOS (Bothnian**
**Sea), BOB (Bay of Bothnia), GOF (Gulf of Finland), GOR (Gulf of Riga).**

|       | BS    | BOB   | BOS   | GOF   | GOR   | KAT   | NBP   | SBP   |
|-------|-------|-------|-------|-------|-------|-------|-------|-------|
| BS    | 1.00  |       |       |       |       |       |       |       |
| BOB   | -0.38 | 1.00  |       |       |       |       |       |       |
| BOS   | 0.06  | 0.57  | 1.00  |       |       |       |       |       |
| GOF   | 0.11  | 0.01  | -0.08 | 1.00  |       |       |       |       |
| GOR   | 0.13  | -0.03 | -0.07 | -0.09 | 1.00  |       |       |       |
| KAT   | 0.42  | -0.04 | -0.03 | -0.10 | 0.30  | 1.00  |       |       |
| NBP   | 0.78  | -0.64 | -0.22 | 0.28  | -0.17 | 0.01  | 1.00  |       |
| SBP   | 0.88  | -     | -0.25 | -0.08 | 0.13  | 0.21  | 0.79  | 1.00  |

| | 0.65 | | | | | | |
|---|---|---|---|---|---|---|---|

Horizontal transition of the trends from positive to negative points to the three-dimensional structure of the FWC variability field. Therefore, time-depth variations of the FWC in each sub-basin were calculated (Fig. 4). Vertically, in the whole Baltic Sea, FWC is the most variable in the halocline layer and beneath it (Fig. 4a). Vertical distribution of the trends shows the absence of the trend in the upper layer of 50 m but strong negative trend within and below the halocline. Thus, the decrease of the FWC in the whole Baltic Sea is mostly contributed by the drop of the FWC below the upper mixed layer. The variability as well as negative trends are strongest in the southern and the northern Baltic Proper (Fig. 4e, c). Moving further northward and eastward in the Baltic Sea, we can notice that the negative tendencies are present only in the deeper layer (deeper than 50 m) of the Gulf of Finland and the Bothnian Sea (Fig. 4f, d). On the other side, there is a strong positive trend of the FWC in the Bothnian Bay and in the upper 50-meter layer of the Bothnian Sea (Fig. 4b, d). It is relevant to note that a positive tendency in the FWC is seen in the layer of 10–50 m in the Gulf of Finland (Fig. 4f). In the northern Baltic Proper, the trend is absent in the upper layer of 30 m but turns negative in the surface layer of the southern Baltic Proper. In the Gulf of Riga, the variability is low, the trends are negligible and do not show systematic vertical distribution.

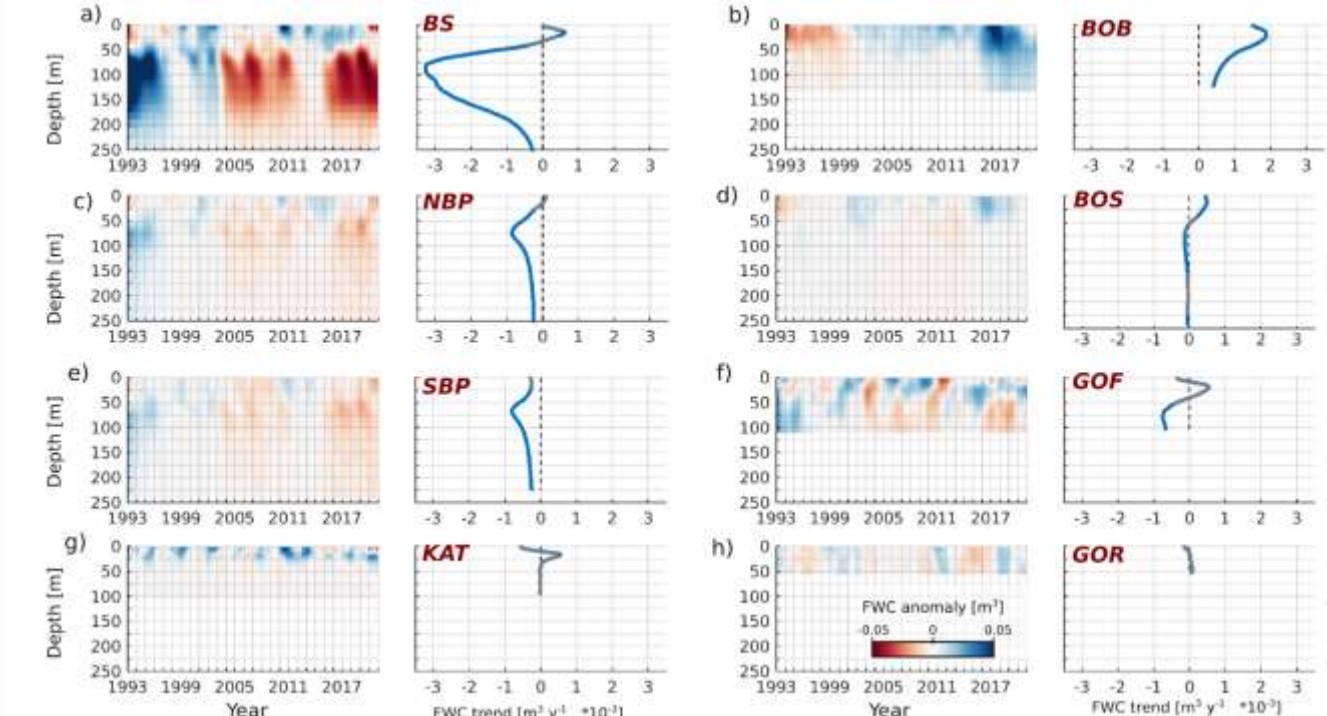

**Figure 4. Vertical distribution of horizontal mean FWC anomaly and corresponding trends for each Baltic Sea sub-basin. Trends with statistical significance less than 5% (p-value <0.05) are shown as grey-shaded. The abbreviations for the sub-basins are as follows: BS (Baltic Sea), NBP (Northern Baltic Proper), SBP (Southern Baltic Proper), KAT (Kattegat), BOB (Bay of Bothnia), BOS (Bothnian Sea), GOF (Gulf of Finland), GOR (Gulf of Riga). (data ref. 1, Table 1)**

FWC exhibits noticeable irregular decadal variations over time, as shown in Fig. 2 and 4. To understand the interannual variability of FWC, we calculated a time series of river runoff, net precipitation, and salt transport through the cross-sections between the sub-basins of the Baltic Sea (Fig. 1; Supplement). Our aim is to examine the co-variability of these factors and the FWC response of both the entire Baltic Sea as well as its sub-basins. To accomplish this, we present a time series of detrended standardised yearly variables on Figure 5.

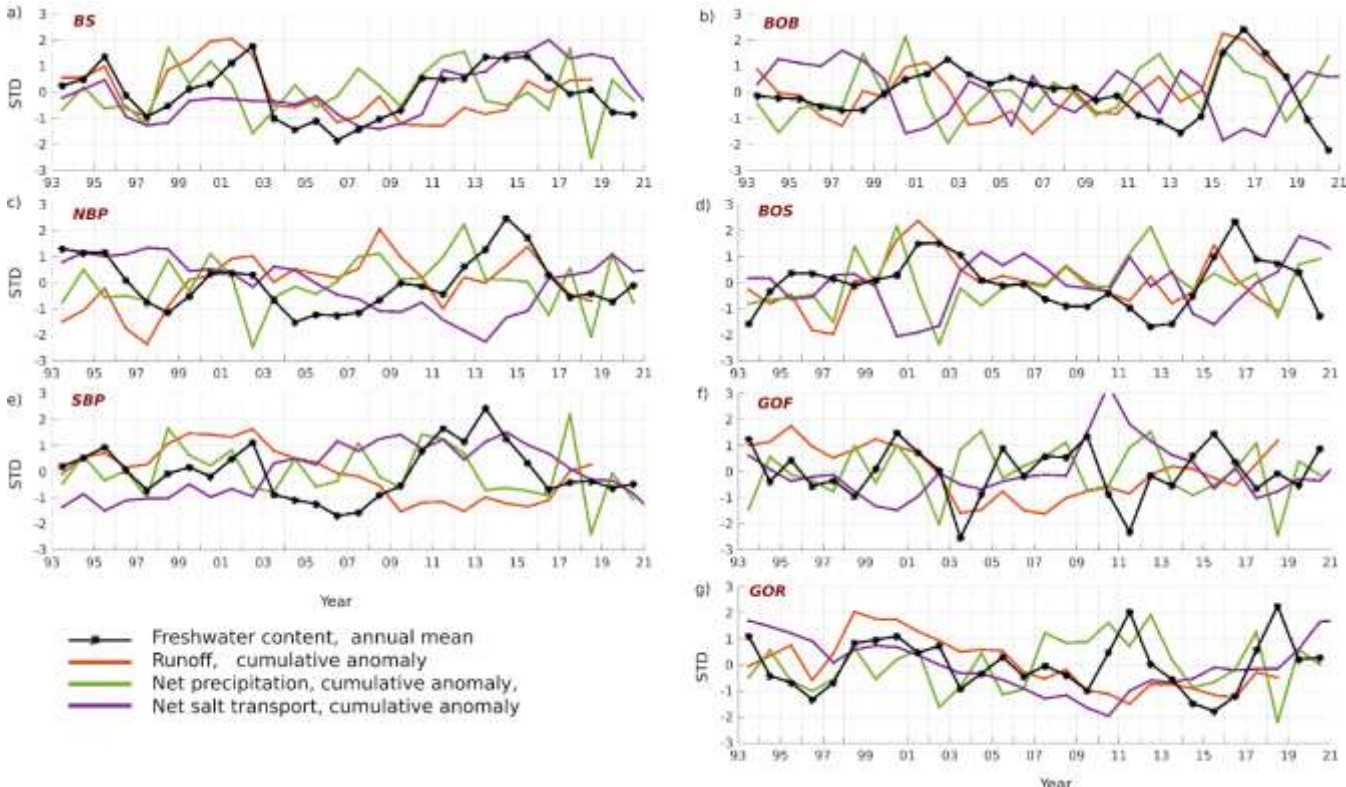

**Figure 5. Normalized time series of detrended annual mean FWC (black asterisk), runoff, net precipitation and net salt fluxes. The abbreviations for the sub-basins are as follows: BS (Baltic Sea), NBP (Northern Baltic Proper), SBP (Southern Baltic Proper), BOB (Bay of Bothnia), BOS (Bothnian Sea), GOF (Gulf of Finland), GOR (Gulf of Riga). (data ref. 1, 2, 3, 1; Table 1)**

For the entire Baltic Sea, the study period can be divided into two sub-periods. From 1993 to 2003, i.e. the first period, changes in the FWC of the Baltic Sea correlated with freshwater sources, namely net precipitation and river runoff (Fig. 5a). A prominent example is the period from 1997 to 2003. Meanwhile, the salt transport to the Baltic Sea remained relatively low. Since 2004, i.e. the second period, we have observed a coherent relationship between salt transport and the FWC in the Baltic Sea, which is difficult to explain. We may speculate that an increase in net precipitation might trigger an increase in the FWC between 2007 and 2012, while river runoff has no effect.

The changes in the FWC of the entire Baltic Sea are influenced by the changes occurring in its sub-basins, resulting in a complex integrated effect. This is illustrated by the simultaneous positive trend of FWC in the northern Baltic Sea and

negative trend in the southern Baltic Sea (Fig. 2) as well as their negative correlation (Table 2). Hence, it is necessary to examine the changes in variables separately for each sub-basin. It is important to note that while the FWC, net precipitation, and river runoff for the entire Baltic Sea represent the sum of contributions from each sub-basin, the net salt flux does not represent an integrated value.

We specifically focus on two sub-basins: the southern Baltic Proper, which is characterised by saltwater dominance, and the Bothnian Bay, which is characterised by freshwater dominance. The variations in the southern Baltic proper differ in several features from the variations in the whole Baltic Sea. The southern Baltic Proper exhibits two distinct periods (Fig. 5e), which is qualitatively similar to the entire Baltic Sea, but the periods do not coincide temporally. The first period spans from 1993 to 2007, while the second period extends from 2008 until the end of recorded data. In the first period, FWC, net

precipitation, and river runoff show positive covariation, while the flux exhibits a negative covariation pattern. In the second period, there is positive covariation between FWC and salt flux and partially net precipitation, but the pattern is reversed for river runoff. In the Bothnian Bay, there is positive covariation between FWC, net precipitation, and river runoff but negative covariation with salt flux (Fig. 5b). In the northern Baltic Proper and the Bothnian Sea, the main pattern is negative covariation between FWC and salt flux (Fig. 5c and 5d). Changes in net precipitation and river runoff generally support

changes in FWC, but the variability pattern remains complex.

The Gulf of Finland (Fig. 5f) and the Gulf of Riga (Fig. 5g) do not exhibit a well-defined pattern in the variability of the variables. In the latter basin, the changes in FWC align with the changes in river runoff from 1993 to 2009 but not thereafter. The seasonal dynamics of FWC further emphasise the effect of freshwater discharge in the northern basins and salt transport in the southern basins of the Baltic Sea (Fig. 6). In the whole Baltic Sea, FWC is low in autumn and winter but high in spring

and summer (Fig. 6a). Qualitatively, low FWC could be explained by high salt transport in autumn and winter (Fig. 7c) accompanied by low river runoff (Fig. 7a), but this is interfered with by high precipitation in autumn (Fig. 7b). Contrarily, the high FWC could be explained by high river runoff and low or even negative salt transport in spring and summer. Indeed, net precipitation is low in that period. The Gulf of Bothnia has low FWC in winter and early spring and high FWC in summer and autumn (Fig. 6b,d). The seasonal course is more pronounced in the Bothnian Bay (Fig. 6b) than in the Bothnian

Sea (Fig. 6d). In the Gulf of Bothnia, the decrease of FWC in winter could be associated with the freezing of seawater (Fig. 7f). An increase of the FWC in the Bothnian Bay in spring coincides with the melting of sea ice (Fig. 7f), high river runoff (Fig. 7a) and negative salt transport (Fig. 7e).

In the southern Baltic proper, FWC is low in winter and high in summer (Fig. 6e), while in the Gulf of Finland the situation is opposite (Fig. 6f). Similarly, net salt transport to these basins is opposite in time (Fig. 7c,d). The seasonal course of FWC

is almost absent in the northern Baltic Proper (Fig. 6c) where the influence from adjacent sub-basins, the southern Baltic Proper and the Gulf of Finland, which have opposite FWC seasonality, could compensate for each other. In the Gulf of Riga, FWC is at its maximum in spring and decreases monotonically until winter (Fig. 6h). Surprisingly, the seasonal course of the FWC in the Kattegat (Fig. 6g) is like the seasonal course of the FWC in the Gulf of Riga (Fig. 6h). Dynamically, these two areas cannot be interlinked due to their geographical separation.

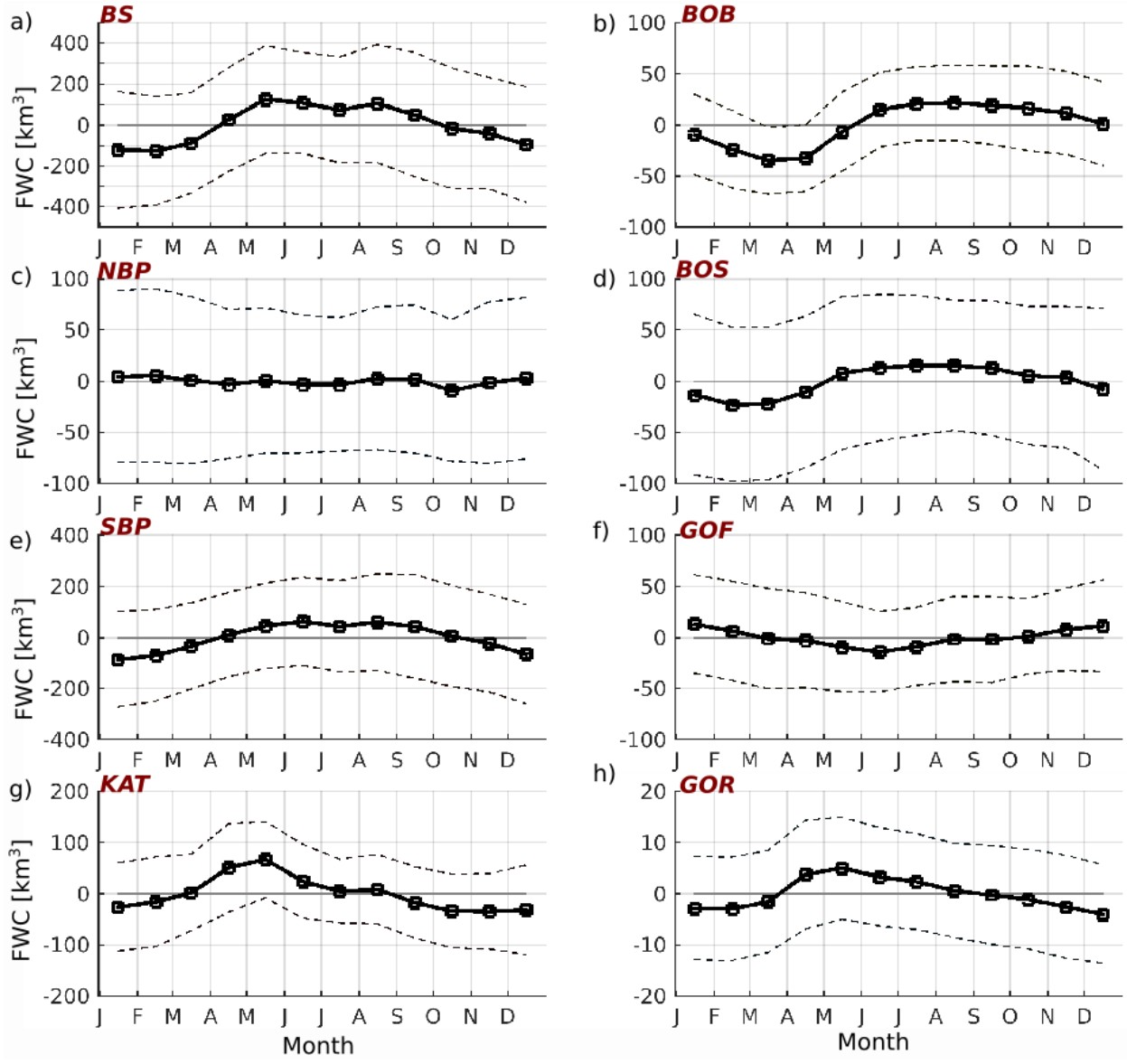

**Figure 6. Seasonality of detrended FWC in the different Baltic Sea sub-basins. The abbreviations for the sub-basins are as follows: BS (Baltic Sea), NBP (Northern Baltic Proper), SBP (Southern Baltic Proper), KAT (Kattegat), BOB (Bay of Bothnia), BOS (Bothnian Sea), GOF (Gulf of Finland), GOR (Gulf of Riga). (data ref. 1, Table 1)**


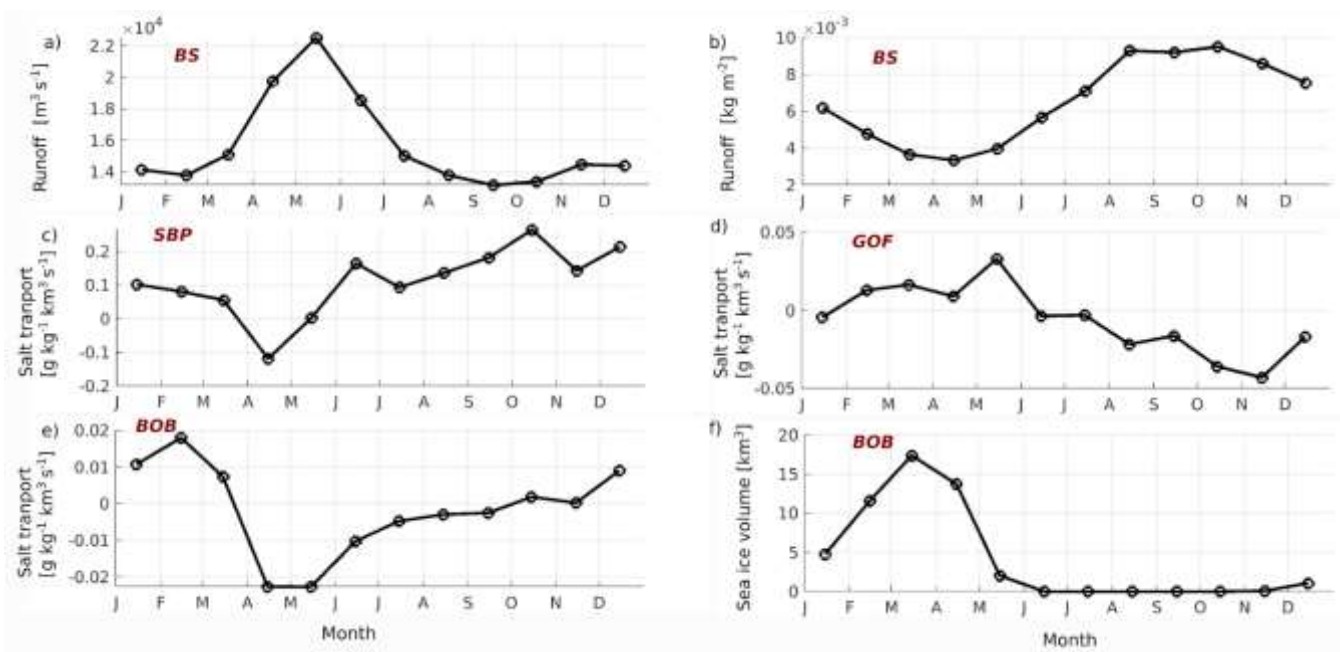

Figure 7. Seasonality of Runoff, net precipitation over the Baltic Sea (BS) (a, b), salt transport through the western boundary of southern Baltic Proper (SBP) (c), the Gulf of Finland (GOF) (d), southern boundary of the Bothnian Bay (BOB) (e), and the sea ice volume in the Bothnian Bay (f).

## 4 Discussion

A distinct feature of the Baltic Sea salinity evolution over the period 1993–2021 was the decreasing trend of the FWC in the southern Baltic Proper and increasing trend in the Bothnian Bay (Fig. 2). Overall, the FWC in the whole Baltic Sea had a statistically significant negative trend (Fig. 2). Salt transport to the Baltic Sea, net precipitation, and total river runoff to the sea (Supplement) did not explain the calculated trends. No steady increase of the salt transport to the Baltic Sea has been reported elsewhere, although deepwater salinity has increased in the Gotland Basin from 1993 to 2018 (Lehmann et al., 2022). The rate of 0.2–0.25 g kg$^{-1}$ per decade was estimated for the period 1979–2018 (Lehmann et al., 2022). In the Baltic Sea, there was no trend in net precipitation nor river runoff. Therefore, the decrease of FWC should be explained by accumulated salt flux from the North Sea to the Baltic Sea due to the Major Baltic Inflows (Mohrholz, 2018), large barotropic inflows (Lehmann et al., 2017), and smaller inflows of barotropic origin (Lehmann et al., 2022). Our estimations of the annual salt transport to the southern Baltic Proper did not show an increase of the salt flux (Supplement). In our reanalysis model data, the salt flux at the Danish straits could have large uncertainty due to relatively high salinity errors there (Appendix A). Inside the Baltic Sea area, as reanalysis model data have been calculated using assimilation of the observations, the estimates of FWC have low uncertainty due to low salinity errors (Appendix A).

Net precipitation and river runoff increased over the Bothnian Bay (Supplement), which contributed to the increase of the FWC (Fig. 2b). An increase of FWC was fastest at the surface of the bay and monotonically slowed down with depth (Fig.

4b). We would like to note that there are no studies that support or refute our findings on the increase of the FWC or the decrease of salinity in the Gulf of Bothnia.

The trends of FWC are present in our data, but this could be characteristic for the period of our study. Increase of the salinity in the central Baltic Proper since 1993 has been reported by Lehmann et al. (2022), which is consistent with our results about the decline of FWC. Going back further in time, the positive salinity trend becomes weaker (Lehmann et al., 2022) until it vanishes (Radtke et al., 2020). The study by Radtke et al. (2020) was prolonged into the past until 1850 but does not cover the last 15 years.

Many authors have reported dominant 30-year variability in mean, surface, and bottom salinity of the Baltic Sea as well as river runoff into the Baltic Sea and even salt transport across Darss sill (Kniebusch et al., 2019; Radtke et al., 2020; Lehmann et al., 2022). Our time series of 28 years are short to reveal 30-year variability in FWC, river runoff, or net precipitation with a statistical significance. Visual inspection of the time series of FWC does not hint to the presence of a 30-year cycle (Fig.2). We admit the presence of decadal variability in the time series, which has been reported also by Lehmann et al. (2022). The trends and multi-scale variability of FWC are opposite in the Baltic Proper and in the Bothnian Bay (Fig. 2, Table 1). This raises the question: If FWC has opposite changes in the southern and northern Baltic Sea, how could changes in river runoff explain the opposite variability. Our analysis showed that there are multi-year periods when river runoff is in phase or out of phase with the FWC. An example of an in-phase period in the whole Baltic Sea is 1993–2009 and an out of phase period 2010–18 (Fig. 5a). In the Bothnian Bay, we mostly have river runoff and the FWC in phase and out of phase with salt transport.

Future climate model scenarios could provide insight into how combinations of different factors affect long term salinity changes in the Baltic Sea, although the uncertainties in the climate projections are high. Three main factors that affect salinity and FWC are the wind fields over the Baltic Sea region (Lass and Matthäus, 1996), river runoff to the Baltic Sea (Schinke and Matthäus, 1998), and global mean sea level rise (Meier et al., 2017, 2021). With increasing precipitation and river runoff, the salinity in the Baltic Sea decreases (Saraiva et al., 2019). Mean sea level rise, in turn, tends to increase salinity because saltwater imports through the Danish straits are larger (Meier et al., 2017, 2021a). An increasing westerly wind could block the freshwater flow out of the Baltic Sea, causing reduced salt transport to the Baltic Sea (Meier and Kauker, 2003), but Schimanke et al. (2014) showed that the intensity and frequency of MBIs were projected to slightly increase due to changes in the wind fields. In summary, different factors affect salinity and FWC differently, so the observed effect cannot be explained in a simple way. For instance, Meier et al. (2021) stated that no changes in the Baltic Sea salinity were found because river runoff and sea level rise approximately compensated each other. Therefore, also in our study, we could not explain the trends and multidecadal variations, although we have considered main factors that affect the FWC of the Baltic Sea.

**Conclusion**

Temporal variability of the FWC at different timescales has an opposite pattern in the northern and southern sub-basins of the Baltic Sea. The Gulf of Bothnia shows positive tendencies of FWC, while in the Baltic Proper, negative tendencies can be witnessed. The total FWC of the Baltic Sea has decreased steadily with a rate of 23.9 km$^3$ y$^{-1}$ over the years 1993–2021. There is no solid explanation for what caused it because different drivers could compensate each other to a certain extent.

Interannual variations of the FWC in the Bothnian Bay are supported by interannual variability of river runoff, net precipitation, and salt transport from the Bothnian Sea. The latter being opposite to the changes of FWC. In the Bothnian Sea and Northern Baltic Proper, interannual variations of the FWC and net salt flux to the basins are opposite to each other. The changes in river runoff and net precipitation have a complex contribution to the changes of FWC. In the Southern Baltic Proper, the changes of FWC, river runoff, net precipitation, and net salt flux have rather complex relationships. A separate study is needed to understand the interplay of these factors, especially seeing as the variations of salt transport and FWC are opposite to each other from 1993 until 2010 but positively correlated thereafter.

The seasonal course of FWC in different sub-basins highlights the local dynamics and explains the FWC dynamics in relation to the local sources of freshwater and salt transport through the sections that border the corresponding sub-basin. Seasonal changes of sea ice volume affect the seasonal cycle of the FWC in the Gulf of Bothnia.

By taking into consideration the spatial and temporal tendencies of the FWC shown in each separate sub-basin, we can characterise the Baltic Sea as a typical estuarine system with a strengthening exchange flow in time. Geographically, the system spans from the Danish straits in the south to the Bothnian Bay in the north. The southern part corresponds to the estuary mouth, where saltwater transport from the ocean prevails and leads to a decrease in FWC. At the other end, the Bothnian Bay is a typical estuary head characterised by a significant influence of freshwater discharge, resulting in an increase in FWC over time. The northern Baltic Proper and the Bothnian Sea converge in the transitional zone between the saltwater-dominated region and the freshwater-dominated region. In terms of vertical distribution, the freshwater-influenced area extends towards the estuary mouth in the upper layer, while the saltwater-dominated area extends towards the estuary head in the lower layer. Moving eastward, the Gulf of Finland represents a branch of the main estuarine system and shares general characteristics of a transition zone, although it possesses its own unique estuarine structure and dynamics (Maljutenko and Raudsepp, 2019; Westerlund et al., 2019; Liblik et al., 2018).

**Appendix A**

We utilize a clustering method to assess the accuracy of the hydrodynamic model. This method provides insights into the overall model accuracy by clustering the errors. The clustering process employs the K-means algorithm, which is a form of unsupervised machine learning (Jain, 2010). The original description of this method can be found in the work of Raudsepp and Maljutenko (2022). In our assessment, all available data within the model domain and simulation period are included,

even if the verification data is unevenly distributed or occasionally sparse. This approach allows us to evaluate the model quality at each specific location and time instance where measurements have been obtained.

The first step of the method is the formation of a two-dimensional error space of two simultaneously measured parameters. A two-dimensional error space $(dS,dT)$, where $dS=(S_{mod}-S_{obs})$ and $dT=(T_{mod}-T_{obs})$, of simultaneously measured temperature and salinity values was formed as the basis for the clustering. The dataset utilized in this validation study was obtained from the EMODNET dataset compiled by SMHI (SMHI, 2019). It comprises a total of 651,565 observations that align with the CMEMS Reanalysis simulation period, encompassing the years 1993 to 2021. We extracted the nearest model values from the reanalysis dataset for each observation.

The second step is the selection of the number of clusters. For simplicity, we preselected five clusters. The third step is to perform a K-means clustering of the 2-dimensional errors. The clustering is applied to the normalized errors. Normalization was done for temperature and salinity errors separately using corresponding standard deviations of the errors. The K-means algorithm finds the location of the centroids of a predefined number of clusters in the error space. The location of the centroids represents the bias of the set of errors for each cluster. The fourth step is calculation of statistical metrics of non-normalized clustered errors. Common statistics, like STD, RMSE, and correlation coefficient can be calculated for the parameters belonging to each cluster.

The fifth step is the analysis of spatio-temporal distributions of the errors belonging to different clusters. In the formation of the error space, we retained the coordinates of each error point $(dS,dT)(x,y)$, which enables us to map the errors belonging to each cluster back to the location where the measurements were performed. In order to do that, the model domain is divided into horizontal grid cells $(i,j)$ of 1x1 km$^2$ in size. Subsequently, the number of error points belonging to different clusters at each grid cell $(i,j)$ is counted. The total number of error points belonging to the grid cell $(i,j)$ is the sum of the points of each cluster. The share of error points in each grid cell belonging to cluster $k$ is the ratio of the number of error points of cluster $k$ and the total number of error points in each grid cell.

Figure A1 displays the results of the K-means clustering for non-normalized errors. Table A1 presents the corresponding metrics. Within cluster $k=5$, the salinity and temperature values closely align with the observations, with a bias of $dS=-0.03$ g kg$^{-1}$ and $dT=0.006$ °C, respectively. This cluster encompasses 77% of all data points. The points are distributed throughout the entire Baltic Sea, with a dominant share exceeding 0.5 (Figure A1b). Clusters $k=3$ and $k=4$ exhibit relatively even spatial distributions over the Baltic Sea, accounting for 8% and 7% of the points, respectively. These clusters are particularly noteworthy due to their small salinity biases and variability, which are crucial for estimating the freshwater content (FWC) directly impacted by salinity. Collectively, approximately 92% of all validation points exhibit relatively low salinity bias, standard deviation (STD), and root mean square error (RMSE) (Table A1). Consequently, we anticipate that the model reanalysis data provide sufficiently accurate information for calculating the FWC of the Baltic Sea.

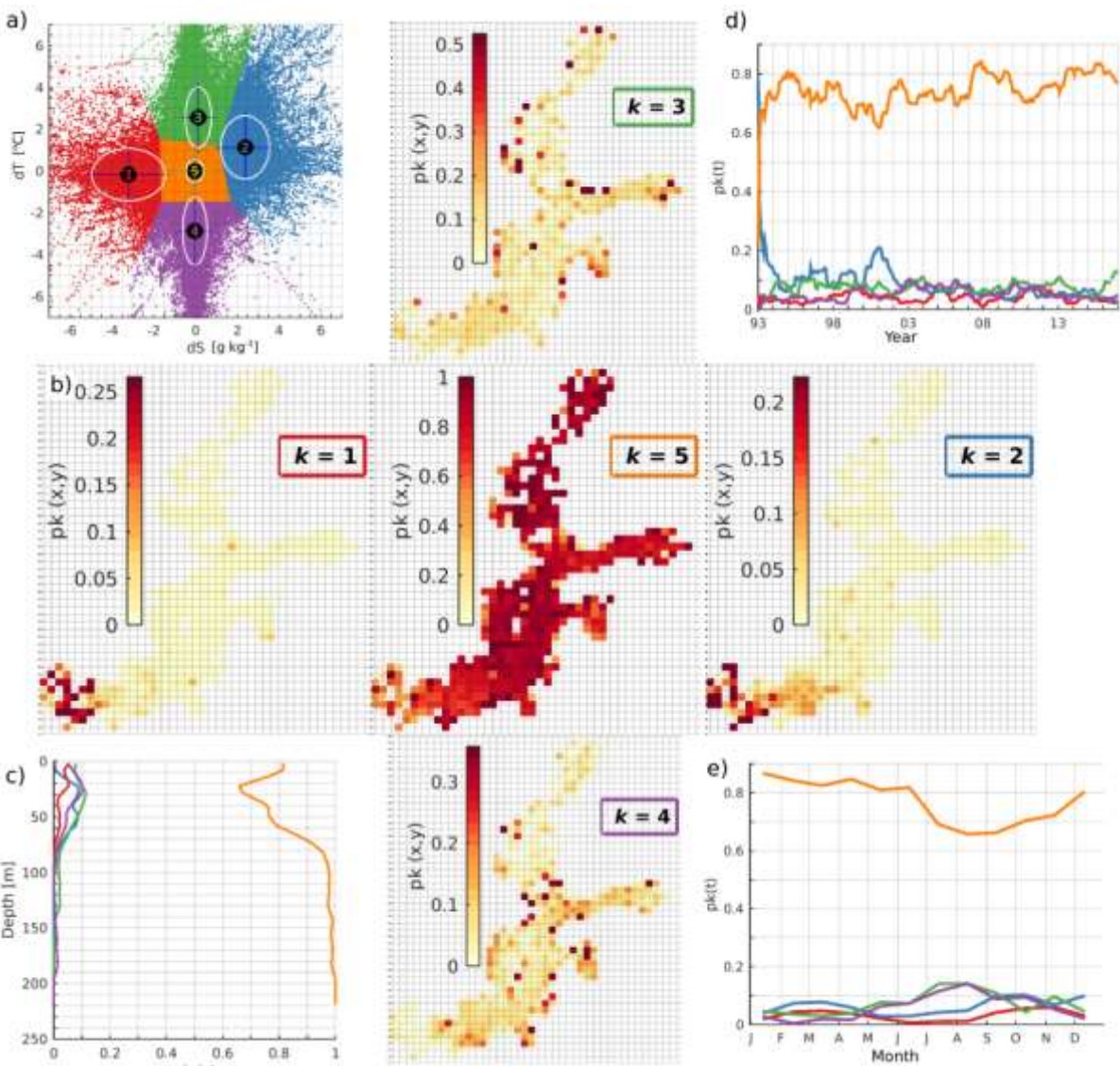

Figure A1. Distribution of normalized error clusters for K=5 (a). The spatial distribution (b, shaded sub-plots), vertical distribution (c), temporal distribution (d), and seasonal distribution (e) of the share of error points belonging to the five different clusters.

**Table A1.** The share (%), bias, root-mean-square error (RMSE), standard deviation (STD) and correlation coefficient (Corr) for each of the five clusters.

| k | Shares % | BIAS dS [g kg⁻¹] | dT [°C] | STD S [g kg⁻¹] | T [°C] | RMSD S [g kg⁻¹] | T [°C] | CORR S | T | dSdT |
|---|---|---|---|---|---|---|---|---|---|---|
| 1 | 2.8 | -3.201 | -0.169 | 1.763 | 1.250 | 3.654 | 1.261 | 0.950 | 0.721 | -0.155 |
| 2 | 5.5 | 2.379 | 1.140 | 1.164 | 1.508 | 2.649 | 1.891 | 0.983 | 0.626 | 0.206 |
| 3 | 8.0 | 0.137 | 2.567 | 0.621 | 1.465 | 0.636 | 2.955 | 0.994 | 0.637 | 0.030 |
| 4 | 6.7 | -0.026 | -2.859 | 0.587 | 1.630 | 0.587 | 3.291 | 0.985 | 0.693 | -0.006 |
| 5 | 77.0 | -0.029 | 0.006 | 0.412 | 0.549 | 0.413 | 0.549 | 0.994 | 0.907 | 0.113 |

**Appendix B**

Mixing of two water masses ($M_{ref}$, $M_f$) with different salinities ($S_{ref}$, $S_f$) results in the mixture ($M_{ref}+M_f$) with unknown salinity $S$

$$M_{ref}S_{ref} + M_f S_f = (M_{ref} + M_f)S. \tag{B2.1}$$

If one of the water masses, $M_f$, is freshwater with $S_f=0$ and we assume that the densities of the water masses of different salinities have negligible difference, then (B2.1) simplifies to

$$V_{ref}S_{ref} = V_{ref}S + V_f S, \tag{B2.2}$$

where $V_{ref}$ and $V_f$ are the volumes of reference water and freshwater, respectively. The formulation of Boyer et al (2007) for freshwater content (FWC) follows directly from (B2.2), where the volume of freshwater in the total volume of the mixture is

$$V_f = V_{ref}\left(\frac{S_{ref}-S}{S}\right) \tag{B2.3}$$

In the derivation of (B2.3), the volume of the mixture of two water masses is not limited.

The formulation (B2.3) answers the question of how much freshwater ($V_f$) is needed to dilute the water ($V_{ref}$) with salinity $S_{ref}$ to the salinity $S$ if $S<S_{ref}$. The volume of water with $S$ will be ($V_{ref}+V_f$). If $S>S_{ref}$, then (B2.3) shows how much freshwater should be removed from the volume of water ($V_{ref}$) with the salinity $S_{ref}$ to obtain water with the salinity $S$. In this case, the volume of resulting water with $S$ will be ($V_{ref}-V_f$).

We would like to note that the relationship (B2.3) is not a linear relationship between salinity and $V_f$. There are straightforward conclusions that can be drawn from (B2.3). First, to obtain water with a salinity close to zero, infinite amount of freshwater is needed, independent of the $S_{ref}$,

$$\lim_{S\to 0} V_f = \infty.$$

Second, no freshwater is needed to add to the water $V_{ref}$ if the mixture has a salinity equal to $S_{ref}$. Third, if

$\lim\limits_{S \to \infty} V_f = -V_{ref}$,

which means that all water should be removed from the mixture.

In its practical application, the formulation (B2.3) means that fixed volume equal to $V_{ref}$ was initially filled with the water of salinity $S_{ref}$. Then, $V_f$ is the volume of freshwater that was needed to dilute the water to the observed salinity $S$. In this case, the volume of the mixture ($V_{ref}+V_f$) with salinity $S$ is larger than the initial volume $V_{ref}$. This means that the amount of water

in the mixture, $V_f$, with salinity $S$ should be removed from the system after the mixing is complete. If $S>S_{ref}$, then $V_f<0$, and this is the volume of freshwater that was removed from the system to obtain water with the salinity $S$. In such a case, the volume of the mixture ($V_{ref}-V_f$) with salinity $S$ is smaller than the initial volume $V_{ref}$. This means that the amount of water in the mixture, $V_f$ with salinity $S$, should be added to the system.

In the natural system, water volume and salinity are conserved when volume and salt exchange with the outside system is

allowed. First, we assume that the natural system is initially filled with water $V_{ref}$ with salinity $S_{ref}$. Then, if at some time instant we observe that water salinity has decreased to $S$, so that $S<S_{ref}$, then $V_f$ was added to the system either by river runoff or net precipitation. (Without losing generality, we neglect the presence of ice in the system and variations of the water volume of the system.) At the same time, to fix the volume of the natural system, the amount of water $V_f$ with salinity $S$ must be removed from the system due to outflow through the open boundary, for instance. If water salinity has increased to $S$, so

that $S>S_{ref}$, then the amount of freshwater $V_f$ has been removed from the system either by evaporation or outflow through the open boundary. Simultaneously, to conserve the volume of water in the natural system, the amount of water $V_f$ with salinity $S$ must be transported to the system due to inflow from the open boundary. We would like to note that in the current argumentation, the actual inflow/outflow volume and salt transport through the open boundary cannot be estimated from knowing the $V_f$.

The derivation of the formulation by Gustafsson and Stigebrandt (1996) is based on the conservation of salt in mixing of two water masses, i.e. (B2.1) and (B2.2) are valid. Additionally, the assumption behind their formulation is that the water volume of the mixture, $V=V_f+V_{ref}$, is known, while $V_{ref}$ is an unknown volume. Thus, in addition to (B2.2), the unknown volume can be expressed as

$V_{ref} = V - V_f$. $\hspace{4cm}$ (B2.4)

Substituting (B2.4) into (B2.2), we can express FWC as

$V_f = V \dfrac{(S_{ref}-S)}{S_{ref}}$. $\hspace{4cm}$ (B2.5)

The relationship between salinity and $V_f$ is linear in (B2.5). Now for some conclusions that can be drawn from (B2.5). First, if the salinity is $S=0$, then $V_f=V$, i.e. all water in the mixture is freshwater. Second, if $S=S_{ref}$, then $V_f=0$, i.e. there is no freshwater in the mixture. If $S>S_{ref}$, then we obtain that $V_f<0$, which means that a fraction of the water in the mixture

becomes negative. Therefore, in its initial applications (Gustafsson and Stigebrandt, 1996; Eilola and Stigebrandt, 1998; Winsor et al., 2001), the relationship (B2.5) is bounded to zero when $S>S_{ref}$. The argumentation about the dynamics of the

freshwater content is not applicable to the formulation by Gustafsson and Stigebrandt (1996). In the applicability range $S \leq S_{ref}$, (B2.5) can be interpreted as follows: initially, a fixed volume, $V$, is filled in with the water with salinity $S_{ref}$; then, the amount of water $V_f$ with salinity $S_{ref}$ is removed from the fixed volume and replaced with freshwater with volume $V_f$ to obtain

the mixture with salinity $S$.

Technically, (B2.5) can also be used for the calculation of $V_f$ if $S > S_{ref}$. We have calculated FWC for the whole Baltic Sea using (B2.3) and (B2.5) (Figure B1). The difference of the calculated FWC of the Baltic Sea is around 100 km$^3$ and rarely exceeds 200 km$^3$, while the FWC anomaly of the Baltic Sea varies in the range of $\pm 1000$ km$^3$. The estimates used by either formulation are between 10–20%. We would like to point out that FWC calculated by (B2.3) (Boyer et al., 2007) is always

larger than FWC calculated by (B2.5) (Gustafsson and Stigebrandt, 1996)

$$\Delta FWC \equiv (B2.5) - (B2.3) = \frac{-(S-S_{ref})^2}{SS_{ref}}. \qquad \text{(B2.6)}$$

From (B2.6) we can see that the difference in FWC of two formulations increases with the increase of the difference between reference water salinity and the salinity of the mixture.

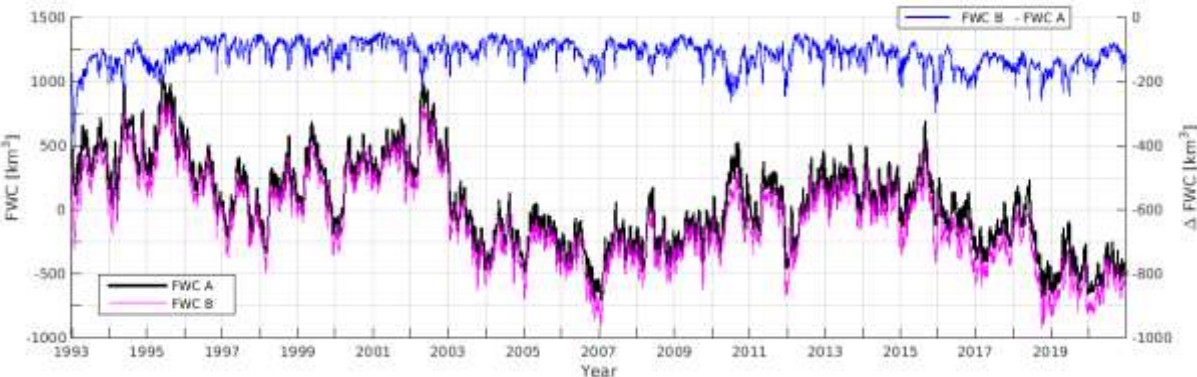

**Figure B1. Time series of FWC in the Baltic Sea as calculated according to (B2.3) (black) and (B2.5) (violet) and difference between (B2.5) and (B2.3) (blue).**

### Data Availability

This study is based on public databases and the references are listed in Table 1.

### Competing Interests

The authors declare that they have no conflict of interest.

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
