# Peer review of "Baltic Sea freshwater content"

_State of the Planet, 2022_

## Community Comment (CC1)

**Reviewer 1**

*Review of « Baltic Sea freshwater content » by Urmas Raudsepp et al.*

*This article is a study of the Baltic Sea freshwater content, based on outputs of re-analyzed models. I think the study is clear and simple, and deserves publication, but there are some points which require further explanation.*

**General comments:**

*You mention several times that the FWC is affected by the sea ice cover. That might be true from a seasonal point of view, but I fail to understand why this would have any effect from an inter-annual point of view since the freshwater stored in sea ice is totally released every year in the water column. That is really the only thing that appears strange for me in this article, and that I think requires further explanation.*

We agree that FWC stored in sea ice is totally released every year. On the other hand, seasonal formation of sea ice affects FWC in the water on an annual scale, if the volume of ice and freshwater stored in the ice is not taken into account in calculation of the FWC in the fixed volume of water. The latter is usually the case in the calculation of the salinity (and FWC) in the ice covered water column. Annual mean FWC is calculated by averaging daily FWC over the year. If the sea ice is formed, then some amount of freshwater is "removed" from the water and "stored" in ice. When the daily volume of ice is larger then more freshwater is stored in the ice. As a consequence, annual mean FWC is smaller when accumulated daily ice volume is larger and vice versa.

In the seasonally ice-covered seas, the ice coverage acts as temporal internal freshwater storage. In a closed water basin without any other sources and sinks, annual mean FWC and accumulated daily ice volume reverse relationship. Therefore our results of the negative trend in annual ice volume and positive trend in FWC in the Bothnian Bay are consistent.

We will provide a detailed explanation of the effect in the revised manuscript.

*Another point, less critical though, is the explanation of the decrease of FWC in the Baltic Proper, which is especially obvious for the deeper parts. You relate this point to an intensification of salt inflows to the Baltic, could you please explicitate what is the scientific consensus, is it related to climate change and/or sea level rise ?*

We are not sure that consensus has reached about the question what has caused intensification of salt transport to the Baltic Sea.

Lehmann et al. (2022) published an overview about the salinity dynamics of the Baltic Sea, where the potential effect of climate change and sea level rise to the salt inflows to the Baltic was discussed. Lehmann et al. (2022) show salinity increase in the deep layer of the Eastern Gotland basin from 1993 until 2018. They add that the frequency of barotropic and major Baltic inflows did not increase during the period. In their overview paper Lehmann et al. (2022) did not give an explanation of the deepwater salinity increase. Also, we do not provide a solid explanation of the decrease of the FWC in the southern Baltic Proper (Eastern Gotland basin is included) (Fig. 2). We show that vertically, decrease of the FWC occurs throughout the water column of the southern Baltic Proper (Fig. 4). We suggest that the most likely reason for the decrease of FWC in the deep layers of the Baltic Sea could be an intensification of salt inflows to the Baltic.

Generally, westerly winds force inflow of saline water and easterly winds force outflow of brackish Baltic Sea water. Over the period 1978-2020, the inflow conditions during months January, February

and March were observed more frequently since the 1990ies (Hindrichen et al., 2022). Thus, if climate change is manifested by an increase of westerly winds in the Baltic Sea region, then an increase of saline water inflows could be resulted.

Hordoir et al. (2015) investigated the influence of sea level rise on saltwater inflows into the Baltic Sea and found an increase in saltwater inflow intensity and frequency with rising sea level (Lehmann et al., 2022). According to Meier et al. (2017) and Saraiva et al. (2019) in future high-end global mean sea level projections, reinforced saltwater inflows result in higher salinity compared to present conditions (Lehmann et al., 2022). Assuming a negligible impact of GMSL rise, the intensity and frequency of MBIs were projected to remain unchanged, with a potential tendency of a slight increase (Schimanke et al., 2014).

One of the key findings of the BACC II assessment was that "Climate model scenarios show a tendency towards future reduced salinity, but due to the large bias in the water balance projections, it is still uncertain whether the Baltic Sea will become less or more saline."

Meier et al. (2022) concluded that "due to the uncertainties in projections of the regional wind, regional precipitation and evaporation, river discharge, and global mean sea level rise, projections of salinity in the Baltic Sea are inherently uncertain, and it remains unknown whether the Baltic Sea will become less or more salty."

We will add discussion about this matter into the revised manuscript.

Lehmann, A., Myrberg, K., Post, P., (...), Lips, U., Bukanova, T., 2022. Salinity dynamics of the Baltic Sea, Earth System Dynamics, 13(1), pp. 373-392.

Markus Meier, H.E., Kniebusch, M., Dieterich, C., (...), Weisse, R., Zhang, W., 2022. Climate change in the Baltic Sea region: A summary, Earth System Dynamics, 13(1), pp. 457-593

Hinrichsen, H.-H., Barz, K., Lehmann, A., Moritz, T., 2022. Can sporadic records of ocean sunfish (Mola mola) in the western Baltic Sea be linked to saline inflow events? Journal of Marine Systems 236,103802. https://doi.org/10.1016/j.jmarsys.2022.103802

Hordoir, R., Axell, L., Löptien, U., Dietze, H., and Kuznetsov, I.: Influence of sea level rise on the dynamics of salt inflows in the Baltic Sea, J. Geophys. Res.-Oceans, 120, 6653–6668, https://doi.org/10.1002/2014JC010642, 2015.

Meier, H. E ., Höglund, A., Almroth-Rosell, E., and Eilola, K.: Impact of accelerated future global mean sea level rise on hypoxia in the Baltic Sea, Clim. Dynam., 49, 163–172, https://doi.org/10.1007/s00382-016-3333-y, 2017.

Saraiva, S., Meier, H. E. M., Andersson, H. C., Höglund, A., Dieterich, C., Gröger, M., Hordoir, R., and Eilola, K.: Uncertainties in projections of the Baltic Sea ecosystem driven by an ensemble of global climate models, Front. Earth Sci., 6, 244, https://doi.org/10.3389/feart.2018.00244, 2019.

Schimanke, S., Dieterich, C., and Meier, H. E. M.: An algorithm based on SLP-fluctuations to identify major Baltic inflow events, Tellus A, 66, 23452, https://doi.org/10.3402/tellusa.v66.23452, 2014.

Radtke, H., Brunnabend, S.-E., Gräwe, U., and Meier, H. E. M.: Investigating interdecadal salinity changes in the Baltic Sea in a 1850–2008 hindcast simulation, Clim. Past, 16, 1617–1642, https://doi.org/10.5194/cp-16-1617-2020, 2020.

Kniebusch, M., Meier, H. E. M., and Radtke, H.: Changing salinity gradients in the Baltic Sea as a consequence of altered freshwater budgets, Geophys. Res. Lett., 46, 9739–9747, 2019b.

**Specific comments:**

*Line 60, can you explicitate the FWC formula so that the reader does not need to read Boyer 2007. Basically my understanding is that your formula is equivalent to saying that*

*FWC= rho(Sref,Tref,p)/rho(0,Tref,p) * (S-Sref)/S*

*which is very different from the usual formula (Sref-S)/Sref since now the variability in time becomes part of the denominator. It would be nice to know what this formula mean, especially why is there a ratio of densities?*

This formula is derived from mixing of two water masses with different salinities by using the conservation of salt:

FWC= -rho(Sref,Tref,p)/rho(0,Tref,p) * (S-Sref)/S                                    (1)

This formula has been used e.g. by Boyer et al. (2007), von Schuckmann et al. (2009), Watelet et al. (2020).

The usual formula FWC~(Sref-S)/Sref is derived from a salt conservation statement also:

FWC= rho(Sref,Tref,p)/rho(0,Tref,p) * (Sref-S)/Sref                          (2)

In several papers, where (2) is used, the reference is given to Boyer et al. (2007), which in our mind creates confusion.

Conceptual difference between the two formulas is as follows. In case of (1) in a fixed volume (Vtot) initially filled in with water Sref, the share of water Sref is replaced by freshwater (VFw) (VSref=Vtot-VFw) and then mixed to obtain defined salinity S. In case of (2) in a fixed volume (Vtot) initially filled in with water Sref, the freshwater VFw is added and two water masses are mixed to obtain the mixture with salinity S. Then a share of mixed water (VFw) is "removed from the system" to conserve the volume Vtot.

Two formulas result in different FWCs. The FWC by (1) is inverse proportional to the water salinity, while FWC by (2) is proportional to water salinity. The FWC anomalies calculated by (1) and (2) for the Baltic are shown in Fig. R1.

The FWC (1) is always greater or equal to FWC (2). Equality of two FWC is achieved when S=Sref. The FWC (1) approaches infinity when S approaches zero.

To clarify the meaning and differences of FWC (1) and FWC (2) we will add Appendix to revised manuscript.

[Figure]

Figure R1. Time series of FWC in the Baltic Sea as calculated according to (1) (black) and (2) (violet) and difference between (2) and (1) (blue).

Boyer, T., Levitus, S., Antonov, J., Locarnini, R., Mishonov, A., Garcia, H. and Josey, S.A., 2007. Changes in freshwater content in the North Atlantic Ocean 1955–2006. Geophysical Research Letters, 34(16).

Von Schuckmann, K., Gaillard, F., Traon, P.-Y.L., 2009. Global hydrographic variability patterns during 2003-2008, Journal of Geophysical Research: Oceans 114(9),C09007.

Watelet, S., Skagseth, Ø., Lien, V.S., (...), Ivshin, V., Beckers, J.-M.,        2020. A volumetric census of the Barents Sea in a changing climate, Earth System Science Data 12(4), pp. 2447-2457.

*Figure 2. It would be nice to have in the figure a reminder of what are BOB, BOS etc. Same for Figure 4.*

We have added the abbreviations into the figure captions. AT - Kattegat, SBP - Southern Baltic Proper, NBP - Northern Baltic Proper, BOS - Bothnian Sea, BOB - Bay of Bothnia, GOF - Gulf of Finland, GOR, Gulf of Riga.

---

## Community Comment (CC2)

**Reviewer 2**

*In this study, freshwater contents of the various sub-basins of the Baltic Sea from the BALMFC CMEMS reanalysis data 1993-2020 were calculated following the method by Boyer et al. (2007). The authors investigated trends in freshwater content per sub-basin and vertically in the water column, as well as seasonal climatologies of freshwater content. In the discussion, trends were attributed to river discharge, net precipitation and sea ice volume changes.*

*Reanalysis data are well suited for the analysis of the ocean conditions and the detection of trends in three dimensions and for the calibration and evaluation of ocean circulation models. However, reanalysis products are generally not good for attribution studies because quantities are not conserved due to the assimilation methods used. As Baltic Sea models often have large biases in salinity due to artificial numerical diffusion (Burchard and Rennau, 2008), data assimilation results in artificial sources and sinks in salinity. Hence, any attribution analysis and discussion of causes of detected changes are difficult.*

We agree with the reviewer that data assimilation does not conserve salt (and heat). Therefore, in this study we described the ocean conditions, i.e. freshwater content of the Baltic Sea and its subbasins. We did not provide salt balance estimation of the Baltic Sea, because the salt balance could be violated due to data assimilation, if the salt transport through the Danish straits is not correctly simulated.

In the revised manuscript, we still will provide more in depth analysis. Keeping in mind that data assimilation is used in the reanalysis product, we will provide analysis of the dynamics and discuss emerging discrepancies and inconsistencies in relation to previous studies.

*Furthermore, the authors considered only river discharge and net precipitation data while wind fields were not analyzed although several previous studies claim that the seasonality in juvenile freshwater propagation or multi-decadal variability in freshwater content are controlled by the wind (the latter at least partly).*

Generally, westerly winds force inflow of saline water and easterly winds force outflow of brackish Baltic Sea water. We agree that juvenile freshwater propagation between the Baltic Sea subbasin is controlled by the wind. In the context of the whole Baltic Sea, the wind fields control saline water inflows to the Baltic Sea, and therefore FWC. In Fig. R2 we plot a time series of fresh water content and the annual accumulation of 10m zonal wind anomaly. In the revised manuscript, we provide transports between the subbasins and their relationships with wind fields.

[Figure]

Figure R2. The time series of fresh water content and the annual accumulation of 10m zonal wind anomaly.

*The provided explanation that melting sea ice could have contributed to the observed positive trends in freshwater content is wrong. In contrast to the Arctic Ocean, in the Baltic Sea multi-year sea ice does not exist. Averaged over one year, the freshwater extraction and freshwater supply is balanced.*

We agree that FWC stored in sea ice is totally released every year. On the other hand, seasonal formation of sea ice affects FWC in the water on an annual scale, if the volume of ice and freshwater stored in the ice is not taken into account in calculation of the FWC in the fixed volume of water. The latter is usually the case in the calculation of the salinity (and FWC) in the ice covered water column. Annual mean FWC is calculated by averaging daily FWC over the year. If the sea ice is formed, then some amount of freshwater is "removed" from the water and "stored" in ice. When the daily volume of ice is larger then more freshwater is stored in the ice. As a consequence, annual mean FWC is smaller when accumulated daily ice volume is larger and vice versa.

In the seasonally ice-covered seas, the ice coverage acts as temporal internal freshwater storage. In a closed water basin without any other sources and sinks, annual mean FWC and accumulated daily ice volume reverse relationship. Therefore our results of the negative trend in annual ice volume and positive trend in FWC in the Bothnian Bay are consistent.

We will provide a detailed explanation of the effect in the revised manuscript.

*The first part of the introduction suggests a relevance of the study for the impact of climate change. However, previous studies found a pronounced multi-decadal variability in salinity and freshwater content of the Baltic Sea (e.g. Winsor et al., 2001). Hence, trends during the rather short period of existing reanalysis data (1993-2020) only describe the natural variability and cannot be used for the analysis of systematic changes.*

The first part of the introduction provides a global background for this study. Our study consists of a time series of almost 30 years. The 30-year period is considered sufficient for climate change studies although longer periods are preferable. We will discuss our results in the context of a multi-decadal variability in salinity and freshwater content of the Baltic Sea (e.g. Winsor et al., 2001; Lehmann et al., 2022).

Lehmann, A., Myrberg, K., Post, P., (…), Lips, U., Bukanova, T., 2022. Salinity dynamics of the Baltic Sea, Earth System Dynamics, 13(1), pp. 373-392.

*Methodologically, the study has gaps. Significance levels of trends are not provided. For me the rationality of the correlation analysis for the understanding of the observed variability is not clear. What have you learned?*

We will provide significant levels of the trends in revised manuscript. We have provided some of the significant levels of trends. The correlation coefficients are calculated because we see similar and opposite changes in the time series of FWC in different basins. Physically, the changes between the subbasins of the Baltic Sea could be correlated. We will provide extended dynamic analysis in the revised manuscript.

*Furthermore, the manuscript suffers from missing references (e.g. Winsor et al. 2001) and phrases that need to be revised (e.g. line 47, line 170).*

We correct the reference list and we will revise the text of the manuscript.

*In summary, the study in the current version is rather descriptive and does not provide any new insights into the causes of observed trends and variability in freshwater content. Hence, I recommend rejection.*

We agree that a major part of the study has been descriptive. The reanalysis data used for the period 1993-2020 and this period is not covered by any of the previous publications. The FWC of the whole Baltic Sea and all main subbasins is described. Previous studies were mainly limited to the central Baltic Sea, either the Gulf of Finland included or not. The seasonal climatology is novel. None of the previous studies have used 3D climatological fields for the reference salinity, but constant values or averages over the shorter periods.

---

## Community Comment (CC3)

**Reviewer 3**

*The research on the fresh water content is a useful approach when addressing estuarine basins. The proposed study aims to make a step towards understanding the evolution of fresh water content in the Baltic Sea based on analysis of BALMFC data. Unfortunately, the manuscript does not provide (1) any validation and error estimates,*

We will add the reanalysis data validation based on the clustering approach by Raudsepp and Maljutenko (2021) in Appendix of the manuscript. In the validation process we include the data that have been used for the assimilation. Argument for that approach is that we like to know how representative the reanalysis product is. Results of the error clustering is shown in Fig. R3 and the statistics are presented in Table R1. Detailed description of the validation procedure and results will be provided in the revised manuscript.

Mutual reanalysis errors of temperature and salinity are used because water temperature is relevant for the ice volume calculations and the salinity for the calculation of the FWC.

[Figure]

[Figure]

Figure R3. The distribution of the error clusters for K=5. The spatial, vertical, temporal, and seasonal (e) distribution of the share of error points belonging to the five different clusters.

Table R1. The bias, root-mean-square error (RMSE), standard deviation (STD) and correlation coefficient (Corr) for each of five clusters.

| | BIAS | | STD | | RMSD | | CORR | | |
|---|---|---|---|---|---|---|---|---|---|
| k | dS | dT | S | T | S | T | S | T | dSdT |
| 1 | -3.201 | -0.169 | 1.763 | 1.250 | 3.654 | 1.261 | 0.950 | 0.721 | -0.155 |
| 2 | 2.379 | 1.140 | 1.164 | 1.508 | 2.649 | 1.891 | 0.983 | 0.626 | 0.206 |
| 3 | 0.137 | 2.567 | 0.621 | 1.465 | 0.636 | 2.955 | 0.994 | 0.637 | 0.030 |
| 4 | -0.026 | -2.859 | 0.587 | 1.630 | 0.587 | 3.291 | 0.985 | 0.693 | -0.006 |
| 5 | -0.029 | 0.006 | 0.412 | 0.549 | 0.413 | 0.549 | 0.994 | 0.907 | 0.113 |

Raudsepp, U., Maljutenko, I., 2022. A method for assessment of the general circulation model quality using K-means clustering algorithm: a case study with GETM v2.5. Geosci. Model Dev., 15, 535–551. doi:10.5194/gmd-15-535-2022

*and (2) sufficient depth of analysis. With the available from BALMFC data, authors could do much more than just a simple diagnostics.*

In the revised manuscript, we will provide more in depth analysis. Still there is possibility that the analysis of the reasons is not consistent with the evolution of the FWC because salt and heat are not necessarily conserved due to data assimilation. That may concern salt transport through the straits in particular. Keeping that in mind we will provide analysis of the dynamics and discuss emerging discrepancies and inconsistencies in relation to previous studies.

*Abstract: „Copernicus regional reanalysis", right term? May be better say Copernicus Baltic Sea regional reanalysis.*

Corrected

*Line 60 big and small s.*

Corrected

*Estimate the errors in the model by analyzing available in situ and satellite data and data from the reanalysis. This would show how credible your results are.*

We will add reanalysis data validation in comparison to in situ data in the revised manuscript as described above.

*The analysis or fresh water content is show-and-tell like. No deep physical explanation of reasons, dynamics etc. are proposed.*

We will provide explanation of reasons and dynamics based on BALMFC reanalysis data used in this study in the revised manuscript.

*It is not always clear whether authors present their own results or results of others. In Line 116-116: "The variability as well as negative trends are strongest in the southern and the northern Baltic*

*Proper (Fig. 4e,c). The decrease of the FWC is explained by the saline water transport from the North Sea to the Baltic Sea by the Major Baltic Inflows (Mohrholz, 2018), large barotropic inflows (Lehmann et al., 2017) and smaller inflows of barotropic origin (Lehmann et al., 2022)." How exactly, what exactly. Use quantitative analyses (BALMFC) to support these statements. Are authors sure that, in BALMFC, the above conclusion hold. This is particularly important for the straits transport. Discuss the realism of straits transports. You may compare with the estimates of Mohrholz et al. (2015), Gräwe et al. (2915), and Stanev et al. (2018). An important question is whether BALMFC correctly represents straits transports.*

We will separate Results and Discussion sections in the revised manuscript and will provide the analysis of the transport between the straits based on the results of BALMFC. This separation will clarify our results and the results of the other.

*Authors mix introduction and results, one example (there are also others): ". Deep layer water in the Gulf of Finland originates from the sub-halocline layer (110–120 m) of the central Baltic Proper (Liblik et al., 2018)." Please, move this in the introduction and restructure your paper. Alternatively, show this from the results of BALMFC, which is preferable. There are in the paper other similar cases. I am not sure whether what a model different from NEMO simulates (or one data set shows) is reproduced by the BALMFC. Support your conclusions with the results from NEMO.*

We will provide an explanation of our results based on BALMFC reanalysis data.

*Much of what is said in the part "Results" cannot be derived from the results: "In winter, the salt wedge withdraws from the interior of the gulf, the mean salinity decreases and FWC increases." Does this follow from BALMFC or other model. Maljutenko and Raudsepp (2019) is not in the reference list.*

We will separate Results and Discussion sections in the revised manuscript and will provide the analysis based on BALMFC reanalysis data. Reference list will be corrected.

*The paragraph starting in line 157 is unclear. Kattegat-Golf of Riga-The whole basin. Author have to try to carefully explain their idea.*

Will be clarified.

*References:*

*Gräwe, U., Naumann, M., Mohrholz, V., Burchard, H., 2015. Anatomizing one of the largest saltwater inflows in the Baltic Sea in December 2014. J. Geophys. Res. 120, 7676–7697*

*Mohrholz, V., Naumann, M., Nausch, G., Krüger, S., Gräwe, U., 2015. Fresh oxygen for the Baltic Sea – an exceptional saline inflow after a decade of stagnation. J. Mar. Sys 2015.*

*Stanev E, Pein J, Grashorn S, Zhang Y, Schrum C, 2018: Dynamics of the Baltic Sea Straits via Numerical Simulation of Exchange Flows, Ocean Modelling 131:40-58*

---

## Author Response (AR1)

**Reviewer 1**

*Review of « Baltic Sea freshwater content » by Urmas Raudsepp et al.*

*This article is a study of the Baltic Sea freshwater content, based on outputs of re-analyzed models. I think the study is clear and simple, and deserves publication, but there are some points which require further explanation.*

**General comments:**

*You mention several times that the FWC is affected by the sea ice cover. That might be true from a seasonal point of view, but I fail to understand why this would have any effect from an inter-annual point of view since the freshwater stored in sea ice is totally released every year in the water column. That is really the only thing that appears strange for me in this article, and that I think requires further explanation.*

We have removed all text that interannual FWC is affected by the sea ice cover and kept the explanation about seasonal cycle only.

We agree that FWC stored in sea ice is totally released every year. On the other hand, seasonal formation of sea ice affects FWC in the water on an annual scale, if the amount of freshwater stored in the ice is not considered in calculation of the FWC. The latter is usually the case in the calculation of the salinity (and FWC) in the ice-covered water column. Annual mean FWC is calculated by averaging daily FWC over the year. If the sea ice is formed, then some amount of freshwater is "removed" from the water and "stored" in ice. When the daily volume of ice is larger than more freshwater is stored in the ice. As a consequence, annual mean FWC is smaller when accumulated daily ice volume is larger and vice versa.

In the seasonally ice-covered seas, the ice coverage acts as temporal internal freshwater storage. In a closed water basin without any other sources and sinks, annual mean FWC and accumulated daily ice volume reverse relationship. Therefore, our results of the negative trend in annual ice volume and positive trend in FWC in the Bothnian Bay are consistent.

*Another point, less critical though, is the explanation of the decrease of FWC in the Baltic Proper, which is especially obvious for the deeper parts. You relate this point to an intensification of salt inflows to the Baltic, could you please explicitate what is the scientific consensus, is it related to climate change and/or sea level rise ?*

We have proposed that salt transport should be the reason, although our data is spread out for confirming this hypothesis. In our study we have explained that salt transport through the straits could have errors, but in overall the salinity in the Baltic Sea is well reproduced by data assimilation into the model.

We are not sure that consensus has reached about the question what has caused intensification of salt transport to the Baltic Sea.

Lehmann et al. (2022) published an overview about the salinity dynamics of the Baltic Sea, where the potential effect of climate change and sea level rise to the salt inflows to the Baltic was discussed. Lehmann et al. (2022) show salinity increases in the deep layer of the Eastern Gotland basin from 1993 until 2018. They add that the frequency of barotropic and major Baltic inflows did not increase during the period. In their overview paper Lehmann et al. (2022) did not explain the deepwater salinity increase. Also, we do not provide a solid explanation why FWC in the southern Baltic Proper has decreased (Eastern Gotland basin is included) (Fig. 2). We show that vertically, decrease of the

FWC occurs throughout the water column of the southern Baltic Proper (Fig. 4). We suggest that the most likely reason for the decrease of FWC in the deep layers of the Baltic Sea could be an intensification of salt inflows to the Baltic.

Generally, westerly winds force inflow of saline water and easterly winds force outflow of brackish Baltic Sea water. Over the period 1978-2020, the inflow conditions during months January, February and March were observed more frequently since the 1990ies (Hindrichen et al., 2022). Thus, if climate change is manifested by an increase of westerly winds in the Baltic Sea region, then an increase of saline water inflows could be resulted.

Hordoir et al. (2015) investigated the influence of sea level rise on saltwater inflows into the Baltic Sea and found an increase in saltwater inflow intensity and frequency with rising sea level (Lehmann et al., 2022). According to Meier et al. (2017) and Saraiva et al. (2019) in future high-end global mean sea level projections, reinforced saltwater inflows result in higher salinity compared to present conditions (Lehmann et al., 2022). Assuming a negligible impact of GMSL rise, the intensity and frequency of MBIs were projected to remain unchanged, with a potential tendency of a slight increase (Schimanke et al., 2014).

One of the key findings of the BACC II assessment was that "Climate model scenarios show a tendency towards future reduced salinity, but due to the large bias in the water balance projections, it is still uncertain whether the Baltic Sea will become less or more saline."

Meier et al. (2022) concluded that "due to the uncertainties in projections of the regional wind, regional precipitation and evaporation, river discharge, and global mean sea level rise, projections of salinity in the Baltic Sea are inherently uncertain, and it remains unknown whether the Baltic Sea will become less or more salty."

We have added discussion about this matter into the revised manuscript.

Lehmann, A., Myrberg, K., Post, P., (...), Lips, U., Bukanova, T., 2022. Salinity dynamics of the Baltic Sea, Earth System Dynamics, 13(1), pp. 373-392.

Markus Meier, H.E., Kniebusch, M., Dieterich, C., (...), Weisse, R., Zhang, W., 2022. Climate change in the Baltic Sea region: A summary, Earth System Dynamics, 13(1), pp. 457-593

Hinrichsen, H.-H., Barz, K., Lehmann, A., Moritz, T., 2022. Can sporadic records of ocean sunfish (Mola mola) in the western Baltic Sea be linked to saline inflow events? Journal of Marine Systems 236,103802. https://doi.org/10.1016/j.jmarsys.2022.103802

Hordoir, R., Axell, L., Löptien, U., Dietze, H., and Kuznetsov, I.: Influence of sea level rise on the dynamics of salt inflows in the Baltic Sea, J. Geophys. Res.-Oceans, 120, 6653–6668, https://doi.org/10.1002/2014JC010642, 2015.

Meier, H. E. ., Höglund, A., Almroth-Rosell, E., and Eilola, K.: Impact of accelerated future global mean sea level rise on hypoxia in the Baltic Sea, Clim. Dynam., 49, 163–172, https://doi.org/10.1007/s00382-016-3333-y, 2017.

Saraiva, S., Meier, H. E. M., Andersson, H. C., Höglund, A., Dieterich, C., Gröger, M., Hordoir, R., and Eilola, K.: Uncertainties in projections of the Baltic Sea ecosystem driven by an ensemble of global climate models, Front. Earth Sci., 6, 244, https://doi.org/10.3389/feart.2018.00244, 2019.

Schimanke, S., Dieterich, C., and Meier, H. E. M.: An algorithm based on SLP-fluctuations to identify major Baltic inflow events, Tellus A, 66, 23452, https://doi.org/10.3402/tellusa.v66.23452, 2014.

Radtke, H., Brunnabend, S.-E., Gräwe, U., and Meier, H. E. M.: Investigating interdecadal salinity changes in the Baltic Sea in a 1850–2008 hindcast simulation, Clim. Past, 16, 1617–1642, https://doi.org/10.5194/cp-16-1617-2020, 2020.

Kniebusch, M., Meier, H. E. M., and Radtke, H.: Changing salinity gradients in the Baltic Sea as a consequence of altered freshwater budgets, Geophys. Res. Lett., 46, 9739–9747, 2019b.

**Specific comments:**

*Line 60, can you explicitate the FWC formula so that the reader does not need to read Boyer 2007. Basically my understanding is that your formula is equivalent to saying that*

*FWC= rho(Sref,Tref,p)/rho(0,Tref,p) * (S-Sref)/S*

*which is very different from the usual formula (Sref-S)/Sref since now the variability in time becomes part of the denominator. It would be nice to know what this formula mean, especially why is there a ratio of densities?*

We have added Appendix B, where we explain the differences of two formulations. A ratio of densities comes from the conservation of mass on which the derivation of the formula is based.

*Figure 2. It would be nice to have in the figure a reminder of what are BOB, BOS etc. Same for Figure 4.*

We have added the abbreviations into the figure captions. KAT (Kattegat), SBP (Southern Baltic Proper), NBP (Northern Baltic Proper), BOS (Bothnian Sea), BOB (Bay of Bothnia), GOF (Gulf of Finland), GOR (Gulf of Riga).

**Reviewer 2**

*In this study, freshwater contents of the various sub-basins of the Baltic Sea from the BALMFC CMEMS reanalysis data 1993-2020 were calculated following the method by Boyer et al. (2007). The authors investigated trends in freshwater content per sub-basin and vertically in the water column, as well as seasonal climatologies of freshwater content. In the discussion, trends were attributed to river discharge, net precipitation and sea ice volume changes.*

*Reanalysis data are well suited for the analysis of the ocean conditions and the detection of trends in three dimensions and for the calibration and evaluation of ocean circulation models. However, reanalysis products are generally not good for attribution studies because quantities are not conserved due to the assimilation methods used. As Baltic Sea models often have large biases in salinity due to artificial numerical diffusion (Burchard and Rennau, 2008), data assimilation results in artificial sources and sinks in salinity. Hence, any attribution analysis and discussion of causes of detected changes are difficult.*

We agree with the reviewer that data assimilation does not conserve salt (and heat). Therefore, in this study we described the ocean conditions, i.e. freshwater content of the Baltic Sea and its subbasins. We did not provide salt balance estimation of the Baltic Sea, because the salt balance could be violated due to data assimilation if the salt transport through the Danish straits is not correctly simulated.

In the revised manuscript, we provided more in-depth analysis. Keeping in mind that data assimilation is used in the reanalysis product. We provided analysis of the dynamics and discussed emerging discrepancies and inconsistencies in relation to previous studies.

*Furthermore, the authors considered only river discharge and net precipitation data while wind fields were not analyzed although several previous studies claim that the seasonality in juvenile freshwater propagation or multi-decadal variability in freshwater content are controlled by the wind (the latter at least partly).*

Generally, westerly winds force inflow of saline water and easterly winds force outflow of brackish Baltic Sea water. We agree that juvenile freshwater propagation between the Baltic Sea subbasin is controlled by the wind. In the context of the whole Baltic Sea, the wind fields control saline water inflows to the Baltic Sea, and therefore FWC. In Fig. R2 we plot a time series of FWC and annual accumulation of 10m zonal wind anomaly. In the revised manuscript, we provide transports between the subbasins not wind fields. Salt transports are factors that directly influence FWC, while wind fields are the factors that influence salt transport.

[Figure]

Figure R2. The time series of fresh water content in the Baltic Sea and the annual accumulation of 10m zonal wind anomaly.

*The provided explanation that melting sea ice could have contributed to the observed positive trends in freshwater content is wrong. In contrast to the Arctic Ocean, in the Baltic Sea multi-year sea ice does not exist. Averaged over one year, the freshwater extraction and freshwater supply is balanced.*

We have removed all discussion about the role of sea ice in the changes of FWC in interannual timescale.

We agree that FWC stored in sea ice is totally released every year. On the other hand, seasonal formation of sea ice affects FWC in the water on an annual scale, if the volume of ice and freshwater stored in the ice is not considered in calculation of the FWC in the fixed volume of water. The latter is usually the case in the calculation of the salinity (and FWC) in the ice-covered water column. Annual mean FWC is calculated by averaging daily FWC over the year. If the sea ice is formed, then some amount of freshwater is "removed" from the water and "stored" in ice. When the daily volume of ice is large then more freshwater is stored in the ice. Consequently, annual mean FWC is smaller when accumulated daily ice volume is larger and vice versa.

In the seasonally ice-covered seas, the ice coverage acts as temporal internal freshwater storage. In a closed water basin without any other sources and sinks, annual mean FWC and accumulated daily ice volume have reverse relationship. Therefore, our results of the negative trend in annual ice volume and positive trend in FWC in the Bothnian Bay are consistent.

We have dropped all discussion about the effect of the sea ice volume in in the revised manuscript.

*The first part of the introduction suggests a relevance of the study for the impact of climate change. However, previous studies found a pronounced multi-decadal variability in salinity and freshwater content of the Baltic Sea (e.g. Winsor et al., 2001). Hence, trends during the rather short period of existing reanalysis data (1993-2020) only describe the natural variability and cannot be used for the analysis of systematic changes.*

The first part of the introduction provides a global background for this study. Our study consists of a time series of almost 30 years. The 30-year period is considered sufficient for climate change studies although longer periods are preferable. We will discuss our results in the context of a multi-decadal variability in salinity and freshwater content of the Baltic Sea (e.g. Winsor et al., 2001; Lehmann et al., 2022).

Lehmann, A., Myrberg, K., Post, P., (...), Lips, U., Bukanova, T., 2022. Salinity dynamics of the Baltic Sea, Earth System Dynamics, 13(1), pp. 373-392.

*Methodologically, the study has gaps. Significance levels of trends are not provided. For me the rationality of the correlation analysis for the understanding of the observed variability is not clear. What have you learned?*

The trends with significant levels (p<0.05) are presented on the plots with asterisk. The correlation coefficients are calculated because we see similar and opposite changes in the time series of FWC in different basins. Physically, the changes between the subbasins of the Baltic Sea could be correlated. We have explained why we have calculated correlation coefficients between FWC time series in the revised manuscript.

*Furthermore, the manuscript suffers from missing references (e.g. Winsor et al. 2001) and phrases that need to be revised (e.g. line 47, line 170).*

We have corrected the reference list.

*In summary, the study in the current version is rather descriptive and does not provide any new insights into the causes of observed trends and variability in freshwater content. Hence, I recommend rejection.*

We agree that a major part of the study has been descriptive. The reanalysis data used for the period 1993-2020 and this period is not covered by any of the previous publications. Furthermore, the Copernicus Marine Service data has not been used for the description of FWC previously.

**Reviewer 3**

*The research on the fresh water content is a useful approach when addressing estuarine basins. The proposed study aims to make a step towards understanding the evolution of fresh water content in the Baltic Sea based on analysis of BALMFC data. Unfortunately, the manuscript does not provide (1) any validation and error estimates,*

We have added reanalysis data validation based on the clustering approach by Raudsepp and Maljutenko (2021) in Appendix A of the manuscript. In the validation process we included the data that have been used for the assimilation. Argument for this approach is that we like to know how representative the reanalysis product is.

Raudsepp, U., Maljutenko, I., 2022. A method for assessment of the general circulation model quality using K-means clustering algorithm: a case study with GETM v2.5. Geosci. Model Dev., 15, 535–551. doi:10.5194/gmd-15-535-2022

*and (2) sufficient depth of analysis. With the available from BALMFC data, authors could do much more than just a simple diagnostics.*

In the revised manuscript, we have provided more in depth analysis using the BALMFC data and another available data.

*Abstract: „Copernicus regional reanalysis", right term? May be better say Copernicus Baltic Sea regional reanalysis.*

Corrected

*Line 60 big and small s.*

Corrected

*Estimate the errors in the model by analyzing available in situ and satellite data and data from the reanalysis. This would show how credible your results are.*

We have added Appendix A about reanalysis data validation in comparison to in situ data.

*The analysis or fresh water content is show-and-tell like. No deep physical explanation of reasons, dynamics etc. are proposed.*

We have provided explanation of the reasons and dynamics.

*It is not always clear whether authors present their own results or results of others. In Line 116-116: "The variability as well as negative trends are strongest in the southern and the northern Baltic Proper (Fig. 4e,c). The decrease of the FWC is explained by the saline water transport from the North Sea to the Baltic Sea by the Major Baltic Inflows (Mohrholz, 2018), large barotropic inflows (Lehmann et al., 2017) and smaller inflows of barotropic origin (Lehmann et al., 2022)." How exactly, what exactly. Use quantitative analyses (BALMFC) to support these statements. Are authors sure that, in BALMFC, the above conclusion hold. This is particularly important for the straits transport. Discuss the realism of straits transports. You may compare with the estimates of Mohrholz et al. (2015), Gräwe et al. (2915), and Stanev et al. (2018). An important question is whether BALMFC correctly represents straits transports.*

We have separated Results and Discussion sections in the revised manuscript. We have calculated salt transport to the Baltic Sea and between the subbasins and presented actual values in the Supplement). This separation clarifies our results and the results of the other.

*Authors mix introduction and results, one example (there are also others): ". Deep layer water in the Gulf of Finland originates from the sub-halocline layer (110–120 m) of the central Baltic Proper (Liblik et al., 2018)." Please, move this in the introduction and restructure your paper. Alternatively, show this from the results of BALMFC, which is preferable. There are in the paper other similar cases. I am not sure whether what a model different from NEMO simulates (or one data set shows) is reproduced by the BALMFC. Support your conclusions with the results from NEMO.*

We have provided an explanation of our results based on BALMFC reanalysis data. We have removed the abovementioned sentence.

*Much of what is said in the part "Results" cannot be derived from the results: "In winter, the salt wedge withdraws from the interior of the gulf, the mean salinity decreases and FWC increases." Does this follow from BALMFC or other model. Maljutenko and Raudsepp (2019) is not in the reference list.*

We have separated Results and Discussion sections in the revised manuscript and provided the analysis based on BALMFC reanalysis data. Reference list is corrected. We have removed the abovementioned sentence.

*The paragraph starting in line 157 is unclear. Kattegat-Golf of Riga-The whole basin. Author have to try to carefully explain their idea.*

The statement is clarified.

*References:*

*Gräwe, U., Naumann, M., Mohrholz, V., Burchard, H., 2015. Anatomizing one of the largest saltwater inflows in the Baltic Sea in December 2014. J. Geophys. Res. 120, 7676–7697*

*Mohrholz, V., Naumann, M., Nausch, G., Krüger, S., Gräwe, U., 2015. Fresh oxygen for the Baltic Sea – an exceptional saline inflow after a decade of stagnation. J. Mar. Sys 2015.*

*Stanev E, Pein J, Grashorn S, Zhang Y, Schrum C, 2018: Dynamics of the Baltic Sea Straits via Numerical Simulation of Exchange Flows, Ocean Modelling 131:40-58*